# Public R&D Projects-Based Investment and Collaboration Framework for an Overarching South Korean National Strategy of Personalized Medicine

**DOI:** 10.3390/ijerph19031291

**Published:** 2022-01-24

**Authors:** Doyeon Lee, Keunhwan Kim

**Affiliations:** Center for R&D Investment and Strategy Research, Division of Data Analysis, Korea Institute of Science and Technology Information (KISTI), 66, Hoegi-ro, Dongdaemun-gu, Seoul 02456, Korea; dylee@kisti.re.kr

**Keywords:** personalized medicine, public R&D project, government investment, collaboration, framework, strategy

## Abstract

Since the South Korean government designated personalized medicine (PM) as a national strategic task in 2016, it has spared no investment to achieve its goals, which were recently accelerated by the COVID-19 pandemic. This study analyzed investment trends in 17 regions and eight technology clusters related to PM, consisting of 5727 public R&D projects worth USD 148.5 million, from 2015 to 2020. We also illustrated the level of investment for different PM-related technology clusters in each region; various research organizations explicitly verified comparable innovation capabilities for all eight technology fields in 17 regions, showing individual differences in technology areas per region. Our framework provided information to allow implementation of two goals: administering successful PM and improving regional equality in public health and healthcare according to technical and organizational levels. This study empirically demonstrates that it can provide a precise overarching innovation scheme with regional, technical, and organizational dimensions to establish collaboration among different stakeholders, thereby creating a foundation for an overarching national PM strategy.

## 1. Introduction

Due to the ongoing healthcare reforms worldwide, cost-containment strategies, and advancement in new technologies and diagnostic tests [1], leading governments have recognized the importance of personalized or precision medicine (PM) as a potent engine driving economic growth. Thus, the United States Precision Medicine Initiative and the European Union’s (EU’s) “Personalised Medicine” (PerMed) were announced in 2015, and the International Consortium for Personalised Medicine (ICPerMed) was announced in 2016 [2,3,4,5,6]. The US Precision Medicine Initiative is a long-term research endeavor to develop a new model of individualized care with a USD 9269 million investment from 2016 to 2020 [7,8] while PerMed and ICPerMed focused on coordinating and fostering research to develop and evaluate PM approaches at the European and international levels with EUR 3.2 billion spent during 2007–2020. After these announcements, other governments, including South Korea, Japan, and China, set out their national PM strategies. The China Precision Medicine Initiative was announced almost exactly one year after the US Precision Medicine Initiative, with an expected investment of more than USD 9 billion for a 15-year research period [2]. China has invested in three important scientific research fields, enabling a deep understanding of the genetics and biological systems of people, data collection and analysis tools, and improved computing power to make discoveries from largescale data [9]. The Japanese government supports the development of PM, as stated in the Japan Revitalization Strategy 2016 and Healthcare Policy Strategy. Japan plans to initially target rare, insurable, infectious, and undiagnosed diseases as well as cancer, dementia, and pharmacogenomics with the accumulated biodata. Second, it aims to target lifestyle-related diseases, such as diabetes and circulatory diseases, and, third, it seeks to promote genomic therapy for depression and dementia [10]. The South Korean government also announced PM as one of its nine National Strategic Projects on 10 August 2016, including building a genome database from more than 100,000 participants; developing a clinical decision-supporting system; and providing PM trial services for lung, stomach, and colon cancer [11]. Such movement in the transformation of medicine and healthcare has been triggered by the enormous global research investment in genomics and molecular biology [12,13].

Major efforts across governments to achieve the goals and ambitions of these national strategic projects highlight a need to improve the effectiveness with which governments foster innovation. These efforts aim to enhance patient outcomes and support the national economic interest against a background of increasing activity and financially constrained situations. Furthermore, PM is setting a new paradigm in the ongoing COVID-19 pandemic [14]. While implementing unprecedented telehealth and remote patient monitoring technologies to allow continuity in-data collection during the pandemic, PM through digital therapeutics provides continuous physiological monitoring and dynamic responses to maintain healthy homeostasis [15,16]. Thus, recent studies have highlighted governments’ endeavors in achieving the goals of PM through effective collaborations with other stakeholders in the academic, public, private, and health sectors at the international and inter-regional levels [5,17,18,19,20,21]. However, most studies have focused on the normative argument that governments should play a core role in encouraging new and more flexible forms of collaboration networks at the local, national, and international levels [13,22]. Prior research has not contributed to mitigating inherent bias of scientists, and experts participated in the public funding decision-making process to set directions of a variety of programs because useful information was not provided. Thus, it is necessary for scientists and experts to provide salient, credible, and legitimate information stemming from a systematic framework for investment and collaborative research strategies of PM, thereby avoiding the potentially biasing influence in decision making [23,24,25,26]. To complement the lack of such framework studies to implement strategies, we adopted our framework [27,28], which could present the information about the current status of and future directions for the overarching South Korean national research and development (R&D) strategy of PM.

### 1.1. Background Study through Literature Review

#### 1.1.1. PM Initiative in South Korea

The South Korean government announced PM as one of its nine National Strategic Projects on 10 August 2016, building its goal for PM as follows: (1) establish a South Korean precision medical cohort of 100,000 volunteers, aligning with international standards that can be linked with overseas precision medical cohort information and a medical information integration system for hospitals and drug developers to jointly use genome, medical, and health information owned by individual institutions; (2) develop a Clinical Decision Supporting System that assists personalized diagnosis and prescription by analyzing bigdata on PM using artificial intelligence (AI) technology; (3) initiate a PM (prevention, diagnosis, and treatment) trial service for three major cancers (lung, stomach, and colorectal) [11]. In November 2018, the Ministry of Health and Welfare launched a health and medical bigdata platform project to accomplish two goals: (1) improve medical quality and healthcare policies and (2) utilize safe and transparent health and medical bigdata. On 22 May 2019, the government integrated the South Korean PM project into the bio-health industry innovation strategy [29]. Therefore, the South Korean government does not seem to have an overarching strategy in terms of harnessing the potential of PM, increasing its positive impact on public health, or managing its impact on the health system [30]. Thus, it is recommended to develop a national strategy for PM by considering a collaboration among government, scientists, and medical professionals [31]. In particular, we should assess the landscape in terms of which government agencies of South Korea have invested in which research fields and the kinds of R&D projects that have been accomplished to organize the collaboration groups in individual research fields.

#### 1.1.2. Main Elements in the Value Chain of Personalized Medicine

The value chain of PM consists of the basic knowledge from genomics, immunology, infectiology, etc.; pre-diagnostic information from health elucidation, inoculation, lifestyle, prevention, etc.; diagnostics from lab tests, omics technologies, imaging, etc.; and treatment from drugs, surgery, therapies, etc. The genome can be regarded as the complete set of genetic information regarding the construction, development, and maintenance of an organism [6]. Thus, analyzing genomic information lays the foundation for the diagnosis, treatment, and prevention of disease.

Omics technologies, referring to a group of technologies that characterize and quantify the various types of biological molecules (e.g., DNA, RNA, proteins, and metabolites) that make up a cell, allow for a more detailed molecular characterization of individuals. Using this information, an individual’s sensitivity and response to a specific disease as well as its severity can be distinguished, and, ultimately, omics analysis can attempt more accurate diagnosis, targeted treatment, and appropriate intervention according to prognosis [32].

Data-intensive technologies such as high-resolution imaging accelerate the technological advancement required for PMs through the utilization of accumulated data. Eventually, more personalized treatment and care of patients can be maximized by converging genomics and many other omics applications with cutting-edge informatics across the widest possible range of clinical areas into healthcare [33]. Namely, the elements of PM involve findings (knowledge) from basic molecular and clinical discoveries and population science; data (informatics) from biomedical and digital technologies, omics technologies, and imaging; and, ultimately, development of clinical applications [21,32,34]. In this study, we investigated PM from three perspectives (see Figure 1): (1) data: bigdata and AI-based analysis platforms (3 sub-clusters); (2) findings: clinical/empirical research for prediction, diagnosis, and therapy (4 sub-clusters); (3) medical and healthcare service: medical and healthcare service for application of PM.

#### 1.1.3. Theories and Empirical Review

Research-funding organizations must often make difficult decisions about resource allocation [35,36]. Theories regarding decision making generally fall into an information-processing approach that traces back to Herbert Simon and the notion of bounded rationality [35,36,37]. Essentially, information processing in relation to human decision making, containing knowledge acquisition and communication among decision makers, is formed (“bounded”) by the limits of individual attention and heuristics [38]. Consequently, considerable skepticism concerning the reliability and legitimacy of peer review [35] and committee and group meetings [39,40] has risen among certain parts of the scientific world, because scientists and experts who participated in decision making have inherent cognitive bias stemming various sources including the specific case (e.g., data, reference materials, and contextual information), environment, culture and experience (e.g., organizational factor, education and training), and human nature (e.g., personal factors, human brain) [26].

As a general principle to mitigate bias, it is necessary to focus solely on the relevant information [23,24,25,26,41]. Furthermore, the information being processed requires three attributes [23]—salience, credibility, and legitimacy—and governments should establish institutional roles for procedures that can provide and exchange timely, coherent, and trustworthy information and for coordination among stakeholders [42].

Recently, new techniques such as AI or machine learning, which improve accuracy, consistency, and fairness, have been adopted to mitigate human biases and decrease the inevitable variability in decision making [25]. Simultaneously, information-shared platforms, such as that provided by the World Health Organization (platform.who.int/data/), have also been introduced to encourage more discussion and interaction with other stakeholders.

For example, the OECD has provided statistics on R&D expenditure of its members for national and international policymakers, and the statistics have enabled them to measure who conducts and who funds R&D and where it takes place, the level and purpose of such activities, and interactions and collaborations between institutions and sectors to evaluate or understand how R&D contributes to economic growth/activities from a broad perspective [41,43].

Some scholars have used the public (R&D) expenditure in part to considerate direction of innovation policies in decision making. Comparing Indonesian innovation capacity developed by increasing R&D expenditure with Asian countries (Singapore, Malaysia, Thailand, Vietnam, South Korea, and China), Aminullah [44] indicated that the capacity has continued to decline such that the government R&D budget is needed to reach a certain level of GDP. In particular, the acquisition of high technical capability of Indonesia emerged as an urgent policy issue, and it must be coordinated with the science and technology policy for economic development by an integrated R&D governance body.

By reviewing the characteristics of mission-oriented programs for innovation policy, Mazzucato [45] suggested that one of the important principals of policymaking is to actively set a direction of change with dynamic debates across different sectors and actors in an economy through ensuring democratic legitimacy. In particular, investigating the trend of public investment by mission-oriented institutes such as the National Institute of Health (NIH) may be the starting point as a deliberate and targeted direction for discussion of public R&D funding.

By comparing the allocation of EU research funds among member states, Begg [46] argued that the directions of the allocated proportions of funding should be revised considering criteria associated with the challenges the EU confronted, and the higher levels of territorial cooperation projects are encouraged to mitigate the adverse effects of internal borders and to provide innovative solutions to common problems such as environmental issues, energy, and healthcare.

Regarding PM, few studies have investigated the status of research investments for policy directions. Nimmesgern et al. [6] addressed challenges for advancing PM in the context of the medical innovation cycle, and the major challenge was identified as the use of omics technologies, demanding a significant continuous investment. Meanwhile, they briefly discussed the breadth of PM projects aided through EU research funding from FP7 (2007–2013) and Horizon 2020 (2014–2020) programs to address a broad range of R&D activities from largescale data collection, omics technologies development, rare disease research, and new diagnostic testing to piloting PM in healthcare. These prime examples of PM projects allowed experts in the PM community to present the research area to be investigated and initiate projects tackling the diverse challenges requiring investment. Furthermore, it is critical to invest continuously in collaboration research across Europe and the world for successful implementation of PM.

In line with this argument, many studies on PM have commonly emphasized that governments should have an important role in promoting various forms of collaboration for R&D systems and supply chains throughout multiple connecting domains, stakeholders, and supplies at the local, national, and international levels [18,20,47,48] to solve clinical and operational issues including lack of electronic health record tools, absence of clinical decision support tools, and demand for integration of test results [13,49].

Conversely, Maughan [50] argued that optimism for PM may lead funding allocation to areas of apparent success to the detriment of other important research areas including diseases associated with poverty and health inequalities in rural areas [38,39]. Thus, the criteria for research funding should follow clinical and population priorities rather than persisting in the apparent promise of the science to further improvement in health and healthcare.

However, these studies on PM have focused on the normative argument without analyzing the trend and/or status of research funding and presenting the evidence-based collaboration approaches. Thus, it is necessary to provide useful R&D funding-related information that may be utilized in decision making for the direction of PM-related research.

### 1.2. Research Purpose and Questions

This study aims to present a useful framework for developing a South Korean PM approach and policies to support its delivery, realizing its benefits for patients. To establish the framework for the national strategy of PM, it is necessary to identify a basis for the segments of the target research domain and investigate each to discuss the direction of future research from the perspective of government funding and collaboration. As suggested by prior studies [27,28,51], this procedure can become the foundation that clarifies the different statuses and research organizations of target fields, thereby improving the quality of decision making for the national R&D planning strategy. As a result, this study provides useful information about detailed R&D activities such as title, scale of funding, research organizations’ name, region, and project manager of target fields. The primary research questions are as follows:Research Question 1-1: What are the expenditures of R&D projects in PM-related fields in which the South Korea government has invested over the past 5 years (2015–2020)?Research Question 1-2: What are the expenditures of R&D projects in PM-related fields in which the South Korea government has invested from a regional perspective?Research Question 2-1: What has been the trend of investment in PM-related fields in which the South Korea government has invested over the past 5 years (2015–2020)?Research Question 2-2: What were the regional portions of the government R&D funding in PM-related technologies?Research Question 3-1: What kinds of organizations (university, industry, research institutes, and hospital) have contributed to PM-related technologies from a viewpoint of regions?Research Question 3-2: From a regional perspective, which organizations may be served as overarching collaborative R&D partners in each PM-related technology?

## 2. Materials and Methods

### 2.1. Data Collection and Preprocessing

In this study, we collected PM-related R&D projects using the National Institute of Science and Technology, which holds information on all national R&D projects in South Korea. The title and abstract of national R&D project data were translated into English. Then, with experts from universities, research institutes, industries, and hospitals, the following keywords and their variants were used to determine search strategies and final data: “precision medicine”, “precise medicine”, “personalized medicine”, “personalised medicine”, “personalized therapy”, “personalised therapy”, “individualized medicine”, “individualized therapy”, “individualised medicine”, “individualised therapy”, “tailored medicine”, “tailored therapy”, “customized medicine”, “customized therapy”, “customised medicine”, “customised therapy”, “precision health”, “targeted treat”, “targeted therapy”, “preventive medicine”, “predictive medicine”, “cohort precision medicine”, and “omics technologies.” The dataset is described in Table 1. The data on 8478 nationally funded PM-related R&D projects from 112 R&D programs between 2015 and 2020 were collected, and the experts examined whether they were associated with PM, bringing the data sample to 5727 projects. After removing projects with missing investment information, the final dataset contained 5647 projects with a value of USD 1408.5 million (Table 2 and Table 3).

To identify the characteristics of these nationally funded R&D projects, we used the All Science Journal Classification (ASJC) model developed through machine learning employing author keywords from approximately 10,000 recent articles in each ASJC field (e.g., 1311: Genetics, 1312: Molecular Biology) in the Scopus database as the feature and the 344 field names of the ASJC codes used to classify journals in Scopus as the labels [27,28]. Three ASJC codes (labels) were attached to each public R&D project, and the probability of relevance of each assigned ASJC code was indicated based on information (feature) from the title and abstract of the R&D project. In addition, a 10% threshold probability was applied to improve the correlation between the assigned ASJC codes and the nationally funded R&D projects. Figure 2 shows a conceptual diagram of this process.

### 2.2. Co-Occurrence Matrix

Based on earlier studies [27,51,52], a co-occurrence technique was employed to determine PM-related research fields. The number of simultaneous appearances of ASJC codes in a project group indicated the relevance of that project. Specifically, the number of times element *i* (from the first list) and element *j* (from the second list) emerged together in the text was provided by the co-occurrence matrix: *i*,*j* = ASJC codes. The more ASJC codes appear in a group, the higher the relevance of the projects that have these ASJC codes.

### 2.3. Clustering and Network Visualization

The network among projects was built based on the number of appearances of ASJC codes in the projects. All nodes in the network were produced by the titles of the research fields present in the ASJC codes, and the font size implied the frequency of the co-occurrence of each ASJC code compared to other codes. Visualizing the network structure enables researchers to understand the relationship between ASJC codes. This study used the modularity-based clustering method from VOSviewer (Leiden University, Leiden, The Netherlands, Version 1.16.15) for its consistency and accuracy of results [53]. The mapping and clustering were calculated by minimizing Equation (1), which describes the clustering algorithm and is explained in a previous study [52,54].
(1)$$V\left(x_{1,}\ldots ,x_{n,}\right)=\underset{i<j}{\sum }\frac{2mc_{ij}}{c_{i}c_{j}}d_{ij}^{2}-\underset{i<j}{\sum }d_{ij}$$
where *n* is the number of nodes in the network, *m* is the number of links in the network, *c_ij_* is the number of links between nodes *i* and *j*, and *c_i_* is the number of nodes *i.*

With respect to $x_{i}$, …, $x_{n}$, $d_{ij}$ is the distance between nodes *i* and *j*. For the mapping, $d_{ij}$ was calculated using the following formula:(2)$$d_{ij}=>\|x_{i}-x_{j}\|=\sqrt{\overset{p}{\underset{k=1}{\sum }}{\left(x_{ik}-x_{jk}\right)}^{2}}$$
where $x_{i}$ is a vector denoting the location of node *i* in a *p*-dimensional map. For the clustering, $d_{ij}$ was calculated using the following formula:(3)dij=0            if xi=xj1γ        if xi≠ xj 
where $x_{i}$ = integer denotes the cluster to which node *i* belongs, *γ* = resolution parameter.

The numbers of the clustering were decided by the resolution parameter (*γ* > 0); the higher the value of the parameter, the larger the number of clusters created. The number of clusters ranged from 1 (*γ* = 0.1) to 8 (*γ* = 0.9). Considering the number and combination of items (i.e., ASJC codes) in individual clusters and the value chain of PM, eight clusters were finally determined.

### 2.4. Defining the PM-Related Research Fields

The PM-associated research fields were defined throughout the experts’ discussion process, which considered both the information about title and content of the projects and the distribution of ASJC codes comprising each cluster. This process enabled experts to reach the conclusion using a variety of information, thereby improving the legitimacy of the process and the perception of it as unbiased and fair [23,24]. To identify potential international collaboration research organizations in some research areas, experts searched in some targeted funding databases (i.e., the Community Research and Development Information Service of the European Commission, RePORT of the NIH in the US, the United Kingdom Research and Innovation of UK, and the Database of Grants-in-Aid for Scientific Research of Japan). The entire process is detailed in Figure 3.

## 3. Results

### 3.1. PM-Related Research Fields of Public R&D Projects

Figure 4 presents the network visualization of PM-related research fields. In this study, the items/nodes were considered research fields (i.e., ASJC codes), and the links were considered co-occurrence links between research fields. The strength/weight of a link was associated with the number of projects in which the two research fields were shown together. The size of label and circle for each research field were influenced by its weight. The characteristics of each research field were identified by the cluster to which it belonged.

Considering the titles and abstracts of the projects, their representative research fields, and the related keywords, the labels of eight clusters were determined as follows:Cluster 1. Bigdata infrastructure for PM (Omics: Omics-bioinformatics based analysis): Research on the establishment of a core infrastructure for PM based on human informatics, including genomics, transcriptomics, proteomics, and metabolomics.Cluster 2. Empirical and clinical studies for PM (Clinical information: Clinical information-based analysis): Research on the system that collects daily life health information, such as pulse and heartrate from wearable devices for personal health management.Cluster 3. Medical and healthcare services (Service: Medical and healthcare services): Research on data infrastructure that allows storing, processing, and analyzing various medical bigdata (genomic information, health and disease information, living environmental information), while collecting and integrating various medical and health sources such as personal, hospital, and government agencies.Cluster 4. Bigdata infrastructure for PM (Smart-health: Smart-health device-based analysis): Research on the development and verification of algorithms that use medical bigdata from various medical and health sources.Cluster 5. Empirical and clinical studies for PM (Drug: Drug discovery, pre-clinical, and clinical studies): Research on the companion diagnosis, molecular diagnosis, pharmacogenomic analysis, early diagnosis, liquid biopsy technology, and pre-clinical/clinical test.Cluster 6. Empirical and clinical studies for PM (Therapies: Targeted therapies): Research on biomarker analysis utilization, diagnostic kit (next-generation sequencing panel, single nucleotide polymorphisms chip, biochip), and AI-based decision-making support.Cluster 7. Bigdata infrastructure for PM (Cohort: Cohort-based clinical data platform): Clinical research on developing personalized treatments including drug prescriptions, medical devices, and treatment programs based on specific genes and environmental factors using medical and health bigdata.Cluster 8. Empirical and clinical studies for PM (Prediction: Prediction and diagnosis): Research on the public health service to promote PM industry through adopting disease genome analysis service, direct to consumer, and decision support system application in the current medical system.

The eight clusters were selected to explain the value chain of the PM, as described in Section 1.1.2 [13,55]. To further study the PM value chain, the eight clusters were grouped into three categories: (1) Bigdata infrastructure for PM (Data): Omics-bioinformatics based analysis (Cluster 1), Smart-health device-based analysis (Cluster 4), and Cohort-based clinical data platform (Cluster 7); (2) Empirical and clinical studies for PM: Clinical information-based analysis (Cluster 2), Drug discovery, pre-clinical, and clinical studies (Cluster 5), Targeted therapies (Cluster 6), and Prediction and diagnosis (Cluster 8); and (3) Medical and healthcare services (Cluster 3). The following subsections present the statuses or trends of the public R&D projects of PM in South Korea from the technology clusters and regions’ perspective.

### 3.2. Status of Government Investment in PM

#### 3.2.1. Status of Public R&D Projects from a Regional Perspective

The South Korean government spent USD 1408.5 million on PM during 2015–2020. Figure 5 shows the present state of the investment in PM in 17 regions of South Korea. As observed, Seoul and Gyeonggi-do were the most-funded regions, making up 45.0% (USD 634.1 million) and 14.9% (USD 210.1 million) of the government investment, respectively. This indicates that the government’s investment was concentrated in the capital region. Subsequently, Daejeon (USD 12.0 million, 12.0%), Chungcheongbuk-do (USD 97.7 million, 6.9%), Ulsan (USD 66.4 million, 4.7%), Daegu (USD 67.1 million, 4.8%), Gangwon-do (USD 32.1 million, 2.3%), Busan (USD 22.9 million, 1.6%), and Gwangju (USD 22.8 million, 1.6%) were funded in descending order. Information on the proportion of regional investments in PM-related research enables various stakeholders to set the direction for considering appropriate government investments for improving the regional capacities.

#### 3.2.2. Status and Trend of Public R&D Projects by Technology Clusters

Comparing the status of investment differences in an R&D domain is important because the appropriateness of portfolio of R&D projects can be evaluated [56]. Therefore, the first step is to classify projects to aid the prioritization process [57]. Figure 6 shows the total amount of national R&D funding for PM in terms of technology clusters and sub-clusters. Bigdata infrastructure for PM made up 62.4% (USD 879.4 million), followed by Empirical and clinical studies for PM (32.3%, USD 455.5 million) and Medical and healthcare services (5.2%, USD 73.6 million). In the bigdata infrastructure for PM, a significant amount of national R&D funding was invested in omics (CLS 1: Omics-bioinformatics based analysis, USD 342.7 million, 24.3%), followed by Smart-health (CLS 4: Smart-health device-based analysis, USD 285.7 million, 20.3%) and Cohort (CLS 7: Cohort-based clinical data platform, USD 251 million, 17.8%). In the empirical and clinical studies for PM, a significant amount of national R&D expenditure was allocated to Clinical information (CLS 2: Clinical information-based analysis, USD 192.2 million, 13.6%), followed by Drugs (CLS 5: Drug discovery, pre-clinical and clinical studies, USD 98.4 million, 7.0%), Prediction (CLS 8: Prediction and diagnosis, USD 87 million, 6.2%), and Therapies (CLS 6: Targeted therapies, USD 78 million, 5.5%). Small amounts of funding were invested in Services (CLS 3: Medical and healthcare services, USD 73.6 million, 5.2%). Since Omics (CLS 1), Smart-health (CLS 4), and Cohort (CLS 7) are considered as the fundamental technology areas for providing customized medical and healthcare services throughout prediction, diagnosis, and therapy, considerable investment in these areas is expected [6,31]. Meanwhile, the small portion of investment in Services (CLS 3) implied that the medical and healthcare services area remains in its infancy.

Table 4 shows the combined annual growth rate (CAGR) of PM-related fields over the past 5 years. The Medical and healthcare services sector was the fastest-growing value chain (28.2%) among other sectors (Bigdata infrastructure for PM: 22.2%; Empirical and clinical studies for PM: 16.7%). From the technology cluster perspective, Cohort (CLS 7) ranked as the fastest-growing technology cluster with investment increasing from USD 14 million in 2015 to USD 64.6 million in 2020 for a CAGR of 35.8%. The second-fastest-growing cluster was Clinical information (CLS 2) with investment increasing from USD 10.2 million in 2015 to USD 35.5 million in 2020 (CAGR: 28.3%). Additionally, the omics (CLS 1) cluster grew with CAGR of 15.9%, reaching USD 79.6 in 2020, and the Drugs (CLS 5) cluster recorded a CAGR of 11.1%, growing from USD 8.4 million in 2015 to USD 14.1 million in 2020. The results of this study suggest that the government intends to enhance its competence in PM-based technologies such as omics, smart-health devices, and medical research-related technologies [6]. In addition, the direction of R&D activities for medical and healthcare services has risen while accomplishing the advancement of PM-related technologies to some degree.

#### 3.2.3. Status of Public R&D Projects According to Technology Clusters and Regions

From the technology clusters and regions perspectives, the status of public R&D projects was investigated to estimate the regional technological competitiveness. As shown in Table 5 of Section 3.2.1, South Korea PM-related research capacities were concentrated in Seoul, Gyeonggi-do, and Daejeon. Seoul in particular received the highest investment in all technology clusters (Omics: USD 158.1 million, Smart-health: USD 122.4 million, Cohort: USD 104.3 million, Clinical information: USD 72.5 million, Drugs: USD 50.4 million, Prediction: USD 44.3 million, Therapies: USD 51.3 million, and Services: USD 30.9 million). Compared to Daejeon, Gyeonggi-do had a comparative advantage in the Omics (USD 60.3 million), Smart-health (USD 52.4 million), Drugs (USD 20.9 million), and Services (USD 17.3 million) clusters. Meanwhile, Daejeon received more government R&D funding than did Gyeonggi-do in the Clinical information (USD 45.8 million) cluster. Gyeonggi-do and Daejeon received similar investments in the Cohort (USD 27.7 million and USD 25.7 million, respectively) and Prediction (USD 15.1 million and USD 10.7 million, respectively) clusters. However, some regions showed relative advantages or potential for growth in specific technological domains. For example, Chungcheongbuk-do acquired relative competitive edges in the Omics and Cohort clusters with USD 46.4 million and USD 38.2 million of investment, respectively. For the Clinical information cluster, Daegu and Ulsan conducted USD 22.5 million and USD 15.6 million worth of R&D, respectively. In the Therapies cluster, Daegu received the second-highest government funding of USD 13.0 million, followed by Seoul (USD 51.3 million). The status map of the 17 regions of South Korea by eight PM-related research fields is illustrated in Figure 7.

#### 3.2.4. Status of Public R&D Projects According to Technology Clusters, Regions, and Organization Types

To present a list of potential collaborative networks in the PM industry, the current status of public R&D projects was reviewed by type of technology cluster, region, and organization (Table 6). Due to the different growth paths, it is difficult to change how a region has grown to acquire a technological capability [13,33]. To effectively transform a region’s growth path, it is necessary to know what types of regional research organizations existed in a new research field. Thus, this study enabled stakeholders to recognize the strengths and weaknesses of research capacities of region organizations.

A regional R&D portfolio is presented in Table 6. It could be presumed what types of organizations in what region have the competitive edge in the eight technology clusters. Looking at the specific amount of funds and investment rankings, industries and hospitals received the most investment in Seoul (USD 141,547 thousand and USD 79,691 thousand, respectively) followed by Gyeonggi-do (USD 68,242 thousand and USD 11,315 thousand, respectively). Research institutes represented the largest funders in Daejeon (USD 103,521 thousand) followed by Seoul (USD 90,408 million) and Gyeonggi-do (USD 66,571 thousand). Universities in Seoul (USD 79,691 thousand) received the most funding, followed by Ulsan (USD 68,242 thousand) and Gyeonggi-do (USD 54,571 thousand). Agencies represented a significant investment in Chungcheongbuk-do (USD 70,519), where the Korea Center for Disease Control and Prevention Agency is located. Notably, in the Services cluster, industry took the lead in Gangwon-do (USD 7360 thousand) over Seoul (USD 6851 thousand). In addition, industry in Gangwon-do played important roles in the Smart-health (USD 6047 thousand) and Prediction (USD 2408 thousand) clusters after Seoul and Gyeonggi-do. In Daegu, research institutes and universities were the second biggest funders in the Clinical information (USD 11,485 thousand) and Therapies (USD 12,848 thousand) clusters. Industry was ranked as the second biggest funder in the Clinical information (USD 7275 thousand) cluster in Busan, whereas universities played an important role in the Cohort (USD 20,186 thousand) and Clinical information (USD 15,632 thousand) clusters in Ulsan. Hospitals in Incheon were ranked as the second biggest funders in the Smart-health (USD 54,995 thousand) cluster.

#### 3.2.5. Potential National Collaborative Partners in R&D Related to Three Targeted Diseases

As mentioned in Section 1.1.3, many countries have implemented PM goals and continued to seek R&D cooperation measures to realize the goals of the PM and to reduce regional inequalities in medical services and public health. National policymakers and/or national R&D program planners need detailed information, such as the R&D status of related technologies, to form a taskforce on policy issues or discover and promote hyper-collaborative R&D across regions.

The central government of South Korea has strongly insisted on a balanced regional development policy by establishing a knowledge network through industry–academic cooperation with R&D institutions in low-innovation areas [58]. As emphasized in the previous study, the government can select organizations with high R&D capabilities and high technological competitiveness in specific technology fields and provide specific information such as technology and research activities. In addition, it can contribute to more rational decision making based on evidence by periodically providing stakeholders with useful information necessary for various decision-making processes in the national R&D strategy establishment process [59].

Here, we provided three examples of targeted diseases such as cancer, brain disease, and chronic disease for PM inter-regional and/or international collaboration. Table 7 shows information on the field of innovation subjects and organizations, the amount of government investment, and the project manager of representative R&D projects for each of the three target diseases.

## 4. Discussion

### Discussion for Collaborative Overarching R&D Strategy on PM

The proposed framework for an overarching national collaborative R&D strategy for PM provides a variety of information to improve the directions of experts’ funding decision making toward realizing the successful implementation of PM and improving regional equality for public medical and healthcare in terms of regional, technological, and organizational dimensions. To demonstrate the utilization of the framework, we established six research questions.

First, the status and trends of government funding in PM-related technologies during 2015–2020 were examined based on RQ1-1 and RQ1-2. This information can allow experts and/or stakeholders to discuss the appropriateness of national R&D investment while diminishing human biases and reducing the inevitable variability in the decision-making process.

Second, the distribution and trends of R&D funding in PM-related technology areas during 2015–2020 were investigated to grasp the technological competitiveness of the PM field from a regional perspective through RQ2-1 and RQ2-2. South Korean central and regional stakeholders can discuss developing collaboration programs to strengthen mutual competitiveness, focusing on regions with high innovation capability in each research field by using PM’s investment information in terms of both eight research fields and 17 regions as a medium.

Finally, focusing on RQ3-1 and RQ3-2, major innovative organizations in each region were divided into universities, industries, research institutes, and hospitals, and their R&D activities and R&D areas were examined. This information can be used as the basis for collaborative R&D partnerships in technology areas at the national and international levels because it contains the objective contents of R&D projects such as research title, project manager, funding scale, and region.

To review the directions for establishing comprehensive national collaboration and implementation policies of PM while mitigating cognitive bias of stakeholders who participated in decision-making process, it should be preceded by creating information about the status of innovative organizations with technology development competitiveness and diagnosis of R&D portfolios of regions in the industry. This may allow the governments to provide better medical and healthcare services to the aging population in rural areas. Therefore, this framework can be used as an empirical analysis tool to revitalize collaborative R&D between central and local governments and provides necessary information as a starting point for R&D support policy to promote balanced development by fostering specialized industries and strengthen public health.

This framework is not only practical for R&D innovation organizations in each region to discover various cooperative partners across the country, but it can also contribute to discovering opportunities in new R&D and business areas and to creating value by expanding their roles and competitiveness. For example, universities and research institutes can discover new research opportunities, expand science and technology infrastructure, and present directions for education and training for employment and commercialization, and hospitals can ultimately implement personalized medical care and find innovative and useful business benefits.

## 5. Conclusions

Since the South Korean government designated PM as a national strategic task in 2016, it has continued to make investments to achieve its goals, which were recently accelerated due to the COVID-19 pandemic. Furthermore, long-term efforts among academia, public, and private and health sectors are required for successful PM implementation [31]. This requires a fine-tuned investment analysis framework that reflects regional variations due to disparities in relevant assets such as human resources, market size, and institutional factors [27].

This study, which began with our previous work, empirically demonstrated that the framework can present accurate cross-regional innovation plans, considering regional, technical, and organizational dimensions to build horizontal and vertical collaborations between different stakeholders, actor networks, and policy actors in different spaces. Therefore, this study makes four important contributions. First, we presented an investment and collaboration framework that provides information on the status and trends of government R&D investments in the technology sector in the target area. Based on previous studies [27,28], data on national research projects, project period, and funding were used. Thus, this information enables the PM’s R&D managers to form strategic collaborations while considering the limited resources, lack of knowledge, and uncertainty about the market in the early PM industry.

The second contribution is that this study showed how to utilize the framework for PM. It provided a comparative analysis of investment levels in government-funded research projects related to PM for each region and other technology clusters in South Korea during 2015–2020 and described various R&D institutions in each cluster and region. By explicitly presenting innovation capabilities in eight distinct technology areas across 17 regions, it not only showed regional differences in individual areas but also proposed a list of organizations of PM services such as cancer, brain diseases, and chronic diseases by region. Seoul has the highest technological prowess compared to other regions, and some regions have superior competencies in specific technological areas, such as Daegu’s clinical information, Gangwon-do’s service, and Ulsan’s cohort. These results showed empirical evidence for the differentiation of regional competitiveness and promoted the discovery of inter-regional cooperative R&D partners for better rural–urban public health services.

Third, COVID-19 not only highlighted the importance of the global common agenda in the medical system but also awakened the international community’s efforts centered on science and technology cooperation are absolutely essential [14,51]. Based on South Korea’s personalized medical technology capabilities and experiences, the framework presented in this study provides basic information to establish and promote differentiated strategies for technology development cooperation with advanced countries according to similar target diseases, purposes, and functions. In the future, it is expected that South Korea will play a role in practical international cooperation through technical support and cooperation networks with developing countries centered on South Korea’s PM strengths and R&D base organizations.

Finally, many studies [35,36,37] pointed out that the cognitive bias of experts stemming from the limits of individual attention and heuristics incurred skepticism concerning the reliability and legitimacy of the decision making, requiring a procedure that can provide useful information for coordinating stakeholders while mitigating the bias and decreasing variability in decision making. The proposed framework contributed to tackling the gap between theories and practices throughout, providing information that has salience, credibility, and legitimacy [23,24]. Especially, this study can reduce the practical barriers posed by lack of information allowed to debate the direction of public R&D funding in the decision-making process, issued by other scholars [6,45,50]. At the same time, the insufficiency of the normative argument that the governments’ core role in forming PM-related collaboration networks at the local, national, and international levels emphasized in previous studies [2,3,4,5,6,10,12,13,19,22] can be improved by this study.

### Limitations and Further Research

Despite these contributions, our study also posed the same limitations that presented challenging questions for future research [27]. Only the data of public R&D projects from the central government were utilized; due to the absence of a database for the R&D expenditures of the 17 local governments, it is currently impossible to integrate the local government-funded project dataset. Thus, it will be necessary to use the proposed framework to accurately understand the status and trends of PM-related technologies when the local government-funded project dataset was developed and assessed by the public. Despite these restrictions, if a consensus on data openness is formed in 18 local governments in the future and a data-sharing plan is prepared, an analysis of detailed R&D perspectives between the central and local governments based on integrated data can be expected. Another limitation was the lack of funding data from other countries, such as the US, EU, and Japan, which could be employed to conduct a comparative analysis for an international research collaboration network among the technology segments, working toward a successful implementation of PM.

## Figures and Tables

**Figure 1 ijerph-19-01291-f001:**
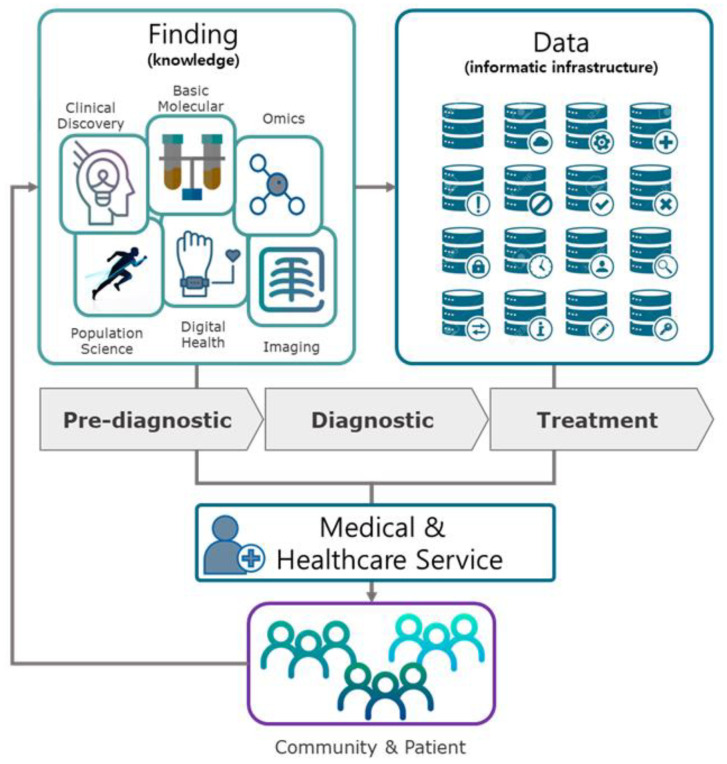
The elements involved in the value chain of PM [21,32,34].

**Figure 2 ijerph-19-01291-f002:**
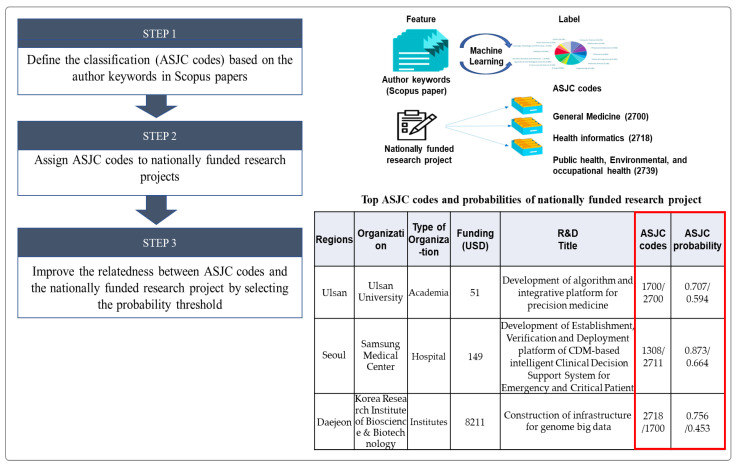
Process of assigning ASJC codes to public R&D projects and improving the correlation between ASJC codes and projects [22].

**Figure 3 ijerph-19-01291-f003:**
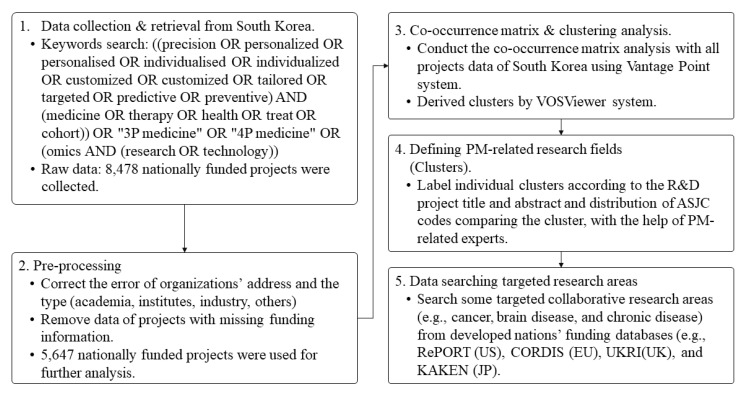
Process of data collection and analysis of nationally funded R&D projects related to PM.

**Figure 4 ijerph-19-01291-f004:**
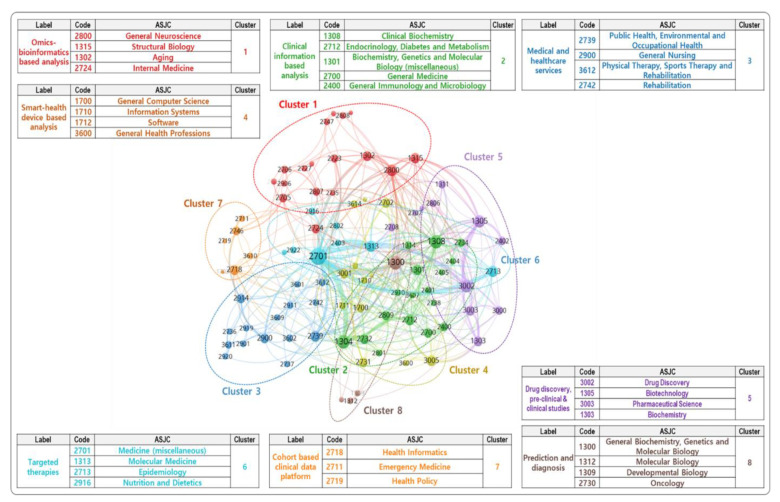
PM-related research fields.

**Figure 5 ijerph-19-01291-f005:**
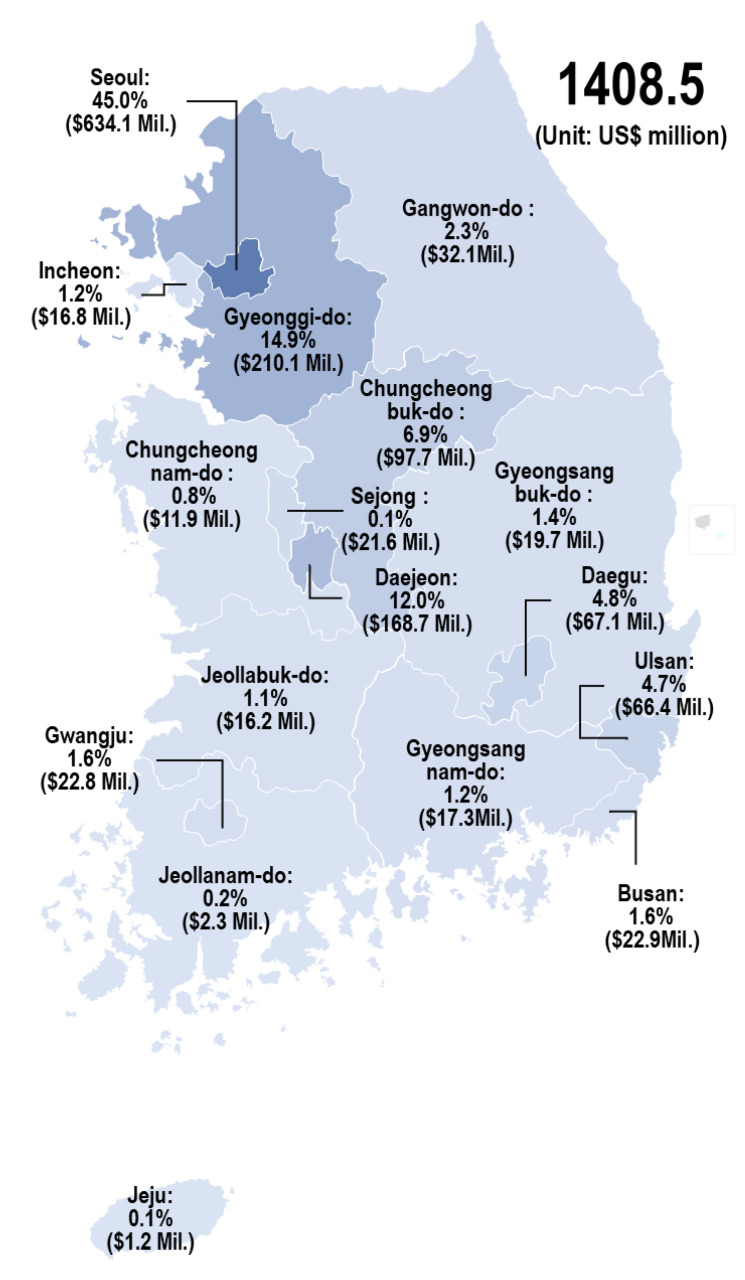
Proportion of PM-related R&D project in the 17 regions of Korea.

**Figure 6 ijerph-19-01291-f006:**
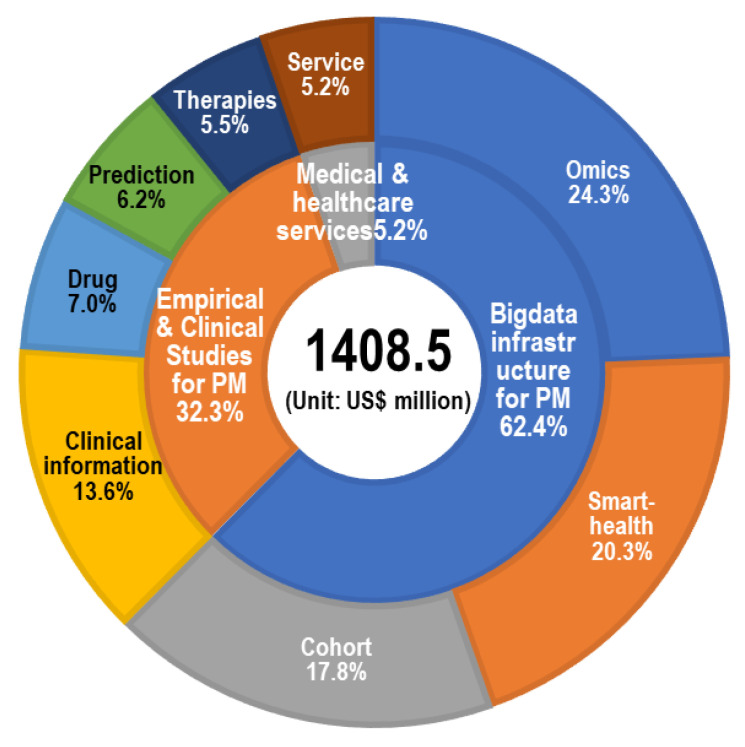
Scale of Korean government investment by technology cluster.

**Figure 7 ijerph-19-01291-f007:**
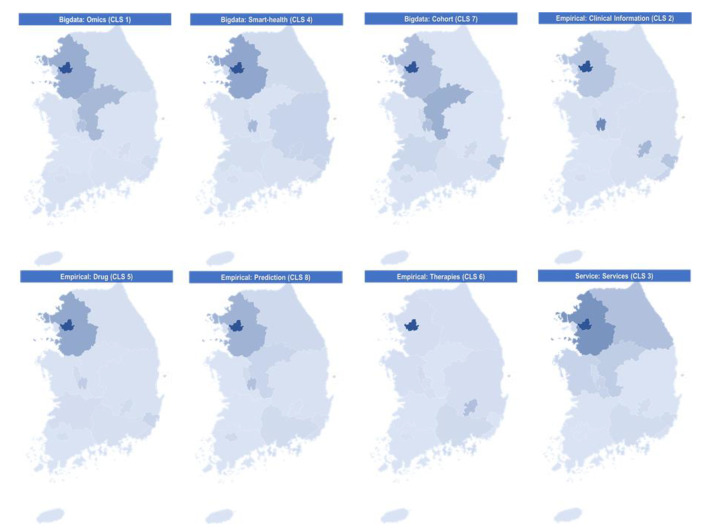
Status maps of the 17 regions of Korea by 8 PM-related research fields.

**Table 1 ijerph-19-01291-t001:** Examples of data on nationally funded R&D projects in the Korea R&D database (NTIS).

Regions	Unique Identification Number (ID)	Organization	Type of Organization	Research Program	Funding (USD Thousand)	Project Period		Project Contents	
						Start Date	End Date	Title	Abstract
Ulsan	1711117189	Ulsan University	University	Omics-based precision medical technology development project	51	9-1-2019	12-31-2024	Development of algorithm and integrative platform for precision medicine	Through the treatment of current biologics available in severe asthma patients using such a treatment response and omics data a new phenotype and cluster through a disturbing effect and select such as to minimize the problem, statistical models developed: PRISM 1 Research. PRISM adaptive design can be applied based on the first results (adaptive design), developing and proposing guidelines for biological agents through clinical tests in the selection of patients with severe asthma: study PRISM 2.
Seoul	1465030239	Samsung Medical Center	Hospital	CDM-based precision medical data integration platform	149	4-17-2019	12-31-2021	Development of Establishment, Verification and Deployment platform of CDM-based intelligent Clinical Decision Support System for Emergency and Critical Patients	First Year (1) Development Goals: General: Consumer emergency center, intensive care CDM extended model development and standardization (based on research) 1 Details: First, Chinese characters CDM extended model standardization and deployment (2) research content and scope (using the system configuration figure, representing the structure, etc.). General (detail 1) research and development information, demand survey carried out in emergency, artificial intelligence algorithms intended for physicians and researchers in the intensive care unit; explore the variables required to build a CDM-based intelligent precision medical identification algorithm.
Daejeon	1711119491	Korea Research Institute of Bioscience and Biotechnology	Research institute	Bio Bigdata	8211	5-29-2020	12-31-2021	Construction of infrastructure for genome big data	(1) Rare, one of the leading business resources and data secure. Holds data of government business resources (leading to business) and a data connection to secure dielectric data (10,000) The dielectric holds leading business (5000) and clinical information (5000) selected by linking genomic data. Rare diseases dielectric secure data (10,000). (2) Creating a dielectric sequencing and analysis report. Leading business and genomic data production of new rare disease samples (15,000).

**Table 2 ijerph-19-01291-t002:** PM-related public R&D projects data and search terms.

Search Terms	Time Period	Amount of Raw Data	Final Number of Data Utilized
((precision OR personalized OR personalised OR individualised OR individualized OR customized OR customized OR tailored OR targeted OR predictive OR preventive) AND (medicine OR therapy OR health OR treat OR cohort)) OR “3P medicine” OR “4P medicine” OR (omics AND (research OR technology))	2015–2020	8478	5647

**Table 3 ijerph-19-01291-t003:** Number of nationally funded R&D projects by research organizations in different regions.

Region	Funding (USD Thousand)	No. of Projects	Funding Per Project	Funding (%)
Gangwon-do	32,145	138	233	2.3%
Gyeonggi-do	210,138	1073	196	14.9%
Gyeongsangnam-do	17,285	95	182	1.2%
Gyeongsangbuk-do	19,722	91	217	1.4%
Gwangju	22,779	131	174	1.6%
Daegu	67,088	163	412	4.8%
Daejeon	168,691	440	383	12.0%
Busan	22,888	125	183	1.6%
Seoul	634,143	2669	238	45.0%
Sejong	1208	8	151	0.1%
Ulsan	66,388	282	235	4.7%
Incheon	16,825	82	205	1.2%
Jeollanam-do	2257	10	226	0.2%
Jeollabuk-do	16,153	64	252	1.1%
Jeju	1199	9	133	0.1%
Chungcheongnam-do	11,908	77	155	0.8%
Chungcheongbuk-do	97,688	190	514	6.9%
Total/Average	1,408,505	5647	249	100.0%

**Table 4 ijerph-19-01291-t004:** Trends of Korean government investment for different technology clusters.

Value Chain Sector	Technology Cluster	2015	2016	2017	2018	2019	2020	Total	2015–2020 CAGR
Bigdata	Omics (CLS 1)	38.1	43.9	54.9	58.3	67.8	79.6	342.7	15.9%
	Smart-health (CLS 4)	21.4	28.6	51.0	71.3	57.6	55.9	285.7	21.2%
	Cohort (CLS 7)	14.0	21.5	29.3	53.1	68.5	64.6	251.0	35.8%
		73.5	94.0	135.2	182.7	193.9	200.1	879.4	22.2%
Empirical	Clinical Information (CLS 2)	10.2	17.5	34.2	51.0	43.7	35.5	192.2	28.3%
	Drug (CLS 5)	8.4	14.6	20.5	21.0	19.8	14.1	98.4	11.1%
	Prediction (CLS 8)	6.3	11.1	13.3	19.1	17.1	20.1	87.0	26.2%
	Therapies (CLS 6)	11.8	15.1	14.2	15.9	11.4	9.6	78.0	−4.1%
		36.7	58.2	82.2	107.0	92.0	79.4	455.5	16.7%
Service	Services (CLS 3)	4.6	8.2	11.9	16.5	16.4	15.9	73.6	28.2%
Total Sum (Unit: USD million)		114.8	160.4	229.3	306.2	302.3	295.5	1408.5	20.8%

**Table 5 ijerph-19-01291-t005:** Status of PM-related research fields in the 17 regions of Korea.

(Unit: USD Million)	Bigdata			Empirical				Service	TOTAL
	Omics (CLS 1)	Smart-Health (CLS 4)	Cohort (CLS 7)	Clinical Information (CLS 2)	Drug (CLS 5)	Prediction (CLS 8)	Therapies (CLS 6)	Service (CLS 3)	
Gangwon-do	7.7	7.9	0.9	2.2	1.5	3.1	1.4	7.5	32.1
Gyeonggi-do	60.3	52.4	27.7	14.7	20.9	15.1	1.7	17.3	210.1
Gyeongsangnam-do	2.1	2.3	2.4	1.7	1.6	2.5	3.2	1.6	17.3
Gyeongsangbuk-do	1.5	12.6	0.2	2.3	0.7	0.5	1.9	0.1	19.7
Gwangju	5.0	2.2	7.7	1.9	2.2	3.1	0.6	-	22.8
Daegu	6.3	12.9	7.5	22.5	2.7	0.6	13.0	1.6	67.1
Daejeon	36.5	37.1	25.7	45.8	7.8	10.7	1.5	3.6	168.7
Busan	3.9	7.6	2.3	7.4	0.5	0.5	0.2	0.6	22.9
Seoul	158.1	122.4	104.3	72.5	50.4	44.3	51.3	30.9	634.1
Sejong	-	-	0.3	-	0.4	0.5	-	-	1.2
Ulsan	9.1	13.1	20.2	15.6	5.2	1.3	0.0	1.8	66.4
Incheon	1.0	7.2	4.0	1.9	0.7	0.3	1.6	0.1	16.8
Jeollanam-do	2.1	0.0	0.0	-	-	-	0.1	-	2.3
Jeollabuk-do	0.1	4.7	9.0	0.1	2.0	-	-	0.3	16.2
Jeju	0.3	-	-	0.2	0.7	-	-	-	1.2
Chungcheongnam-do	2.3	2.4	0.7	1.4	0.3	0.1	1.3	3.4	11.9
Chungcheongbuk-do	46.4	0.9	38.2	2.0	0.8	4.4	0.2	4.9	97.7
Total	342.7	285.7	251.0	192.2	98.4	87.0	78.0	73.6	1408.5

**Table 6 ijerph-19-01291-t006:** The status of public R&D investment by technology cluster and region.

(Unit: USD Thousand)	Organization	Gangwon-do	Gyeonggi-do	Gyeongsangnam-do	Gyeongsangbuk-do	Gwangju	Daegu	Daejeon	Busan	Seoul	Sejong	Ulsan	Incheon	Jeollanam-do	Jeollabuk-do	Jeju	Chungcheongnam-do	Chungcheongbuk-do
Omics (CLS 1)	Industry	-	7299	-	-	-	125	2435	1375	23,593	-	250	528	-	56	-	-	-
	University	7676	14,779	1974	1518	4178	3440	10,197	2490	92,853	-	8874	225	1871	-	250	2269	771
	Hospital	-	498	167	-	807	-	-	-	18,566	-	-	289	214	-	-	-	-
	Institute	-	37,323	-	-	-	2758	23,879	-	22,943	-	-	-	-	-	-	-	15,655
	Agency	-	393	-	-	-	-	-	-	144	-	-	-	-	-	-	-	29,990
Smart-health (CLS 4)	Industry	6047	30,032	1131	2915	825	5475	4125	3262	37,877	-	1164	1301	-	836	-	442	283
	University	1900	12,835	-	2292	613	6961	6360	1475	40,137	-	11,965	308	42	3799	-	1191	25
	Hospital	-	3859	1181	-	-	-	21	-	9467	-	-	5499	-	58	-	83	-
	Institute	-	5654	-	7148	765	58	26,598	-	29,638	-	-	83	-	-	-	656	-
	Agency	-	54	-	231	-	404	-	2863	5244	-	-	-	-	-	-	-	559
Cohort (CLS 7)	Industry	108	2146	1000	148	-	-	3201	1083	28,390	-	-	-	-	4	-	-	1717
	University	747	12,578	242	65	6723	3555	2052	920	55,083	250	20,186	1660	-	7588	-	723	1792
	Hospital	-	4994	1136	-	941	1422	67	334	16,301	42	-	2332	27	1386	-	-	417
	Institute	-	7012	-	-	-	1203	20,417	-	2833	-	-	-	-	-	-	-	-
	Agency	-	929	-	-	-	1271	-	-	1686	-	-	-	-	-	-	-	34,318
Clinical Information (CLS 2)	Industry	375	2563	-	148	-	-	1104	7275	12,925	-	-	-	-	141	-	-	-
	University	1439	4861	1435	2105	1535	8724	12,762	124	43,613	-	15,632	1554	-	-	245	1431	192
	Hospital	376	1388	252	-	388	696	-	-	10,117	-	-	343	-	-	-	-	-
	Institute	-	5908	-	-	-	11,485	18,473	-	5022	-	-	-	-	-	-	-	-
	Agency	-	-	-	-	-	1567	13,447	-	777	-	-	-	-	-	-	-	1770
Drug (CLS 5)	Industry	1499	6599	-	650	-	417	-	100	2767	417	-	73	-	-	704	-	125
	University	-	2785	-	-	613	2300	297	360	17,710	-	5233	592	-	2014	-	292	-
	Hospital	-	167	-	-	1619	-	1200	-	12,538	-	-	-	-	-	-	-	-
	Institute	-	4704	-	-	-	-	6276	-	6532	-	-	-	-	-	-	-	-
	Agency	-	6685	1568	-	-	-	-	-	10,859	-	-	-	-	-	-	-	662
Prediction (CLS 8)	Industry	2408	10,452	-	441	446	-	1904	228	14,521	417	-	292	-	-	-	117	-
	University	667	4287	2190	33	2071	217	4002	273	15,361	83	1310	-	-	-	-	-	3600
	Hospital	-	-	104	-	578	-	21	-	10,129	-	-	-	-	-	-	-	-
	Institute	-	358	-	-	-	417	4763	-	4300	-	-	-	-	-	-	-	-
	Agency	-	-	167	21	42	-	-	-	-	-	-	-	-	-	-	-	783
Therapies (CLS 6)	Industry	-	96	-	-	-	167	1000	-	14,622	-	-	-	-	-	-	56	54
	University	1367	1189	3167	1931	284	12,848	483	167	23,437	-	12	-	-	-	-	675	-
	Hospital	-	204	-	-	353	-	-	-	1642	-	-	1614	103	-	-	-	117
	Institute	-	248	-	-	-	-	-	-	11,617	-	-	-	-	-	-	533	-
	Agency	-	-	-	-	-	-	-	-	-	-	-	-	-	-	-	-	-
(Service (CLS 3)	Industry	7360	9054	-	-	-	-	390	-	6851	-	-	-	-	-	-	392	1906
	University	160	1257	1573	75	-	1025	104	558	10,634	-	1762	117	-	270	-	3049	433
	Hospital	-	206	-	-	-	-	-	-	933	-	-	17	-	-	-	-	81
	Institute	-	5365	-	-	-	-	3115	-	7523	-	-	-	-	-	-	-	-
	Agency	17	1379	-	-	-	555	-	-	4958	-	-	-	-	-	-	-	2437
TOTAL	Industry	17,797	68,242	2131	4302	1271	6183	14,158	13,322	141,547	833	1414	2193	-	1037	704	1006	4085
	University	13,955	54,571	10,579	8020	16,017	39,069	36,257	6368	298,828	333	64,974	4455	1913	13,671	495	9629	6813
	Hospital	376	11,315	2840	-	4685	2117	1308	334	79,691	42	-	10,093	344	1445	-	83	615
	Institute	-	66,571	-	7148	765	15,921	103,521	-	90,408	-	-	83	-	-	-	1190	15,655
	Agency	17	9440	1735	252	42	3797	13,447	2863	23,669	-	-	-	-	-	-	-	70,519

**Table 7 ijerph-19-01291-t007:** Representative collaborative research organizations from university, research institutes, hospital, and industry in cancer, brain disease, and chronic disease of PM.

Target Disease	Type of Organization	Organization	R&D Title	Project Manager	Region	Funding (USD Thousand)
Cancer	Institute	National Cancer Center	Prognostic impact of CT-determined sarcopenia and sarcopenic obesity in older patients with non-small cell lung cancer undergoing chemotherapy	Yoon-jung Jang	Gyeonggi-do	596
	University	Yonsei University	Development of an app-based self-management program “HARU” for cancer patients and testing its effectiveness	Kyungmi Jung	Seoul	11
	University	Seoul National University	Evaluation of risk for oral diseases in cancer patients in Korea and the National Health Insurance coverage extension	Seo-kyung Han	Seoul	75
	University	Yonsei University	Development of prospective cohort and evidence-based management program for colorectal cancer survivors	Seon-ha Ji	Seoul	55
	Institute	Broad Institute Inc.	Making cancer precision medicine real bottlenecks and opportunities	Todd R. Golub	Cambridge, MA, USA	1024
	University	Royal College of Surgeons in Ireland	Advancing a precision medicine paradigm in metastatic colorectal cancer systems-based patient stratification solutions	Annette Byrne PhD	Dublin, Ireland	6794
	University	Queen Mary University of London	Optimal screening and surveillance regimes for early diagnosis of cancer and precision medicine using mathematical modelling	Kit Curtius	London, UK	370
	University	Keio University	Establishment of small cell lung cancer organoids for development of precision medicine	Mitsuishi Akifumi	Tokyo, Japan	37
Brain disease	Hospital	Samsung Medical Center	Protocol development and validation of personalized CNS-PNS hybrid rehabilitation therapy for restoration of gait-related neural network in stroke Patients	Yeon-hee Kim	Seoul	155
	Hospital	Seoul National University Hospital	Modeling of prognosis prediction for stroke using big data	Byung-Woo Yoon	Seoul	108
	Institute	Korea Institute of Science and Technology	Development of customized rehabilitation technology for stroke patients in neural plasticity evaluation and enhancement	In-chan Yoon	Seoul	1083
	University	Pusan National University	Effect of digital treatment system on upper limb functional recovery and brain plasticity in stroke patients	Yong-il Shin	Busan	83
	University	Gachon University	Development of biomarker monitoring system for verification of Korean medicine treatment towards stroke	Young-jun Kim	Gyeonggi	183
	University	Ohio State University	Laying the groundwork for personalized medicine in aphasia therapy genetic and cognitive predictors of restorative treatment response	Stacy M. Harnish	Columbus, Ohio, USA	487
	University	Charité-Universitätsmedizin Berlin	Personalised medicine by predictive modeling in stroke for better quality of life	Dietmar Frey	Berlin, Germany	6773
	University	King’s College London	Towards personalised medicine in psychiatric genetics the role of cardiometabolic traits in severe mental illness	Saskia Hagenaars	London, UK	409
	University	Hamamatsu University School of Medicine	Precision medicine in developmental psychiatry	Kenji J. Tsuchiya	Shizuoka, Japan	159
Chronic disease	Industry	M2IT	Intelligent diagnosis prescription inquiry service using CDM-based chronic disease data	Wooseop Shin	Seoul	417
	Agency	Korea Disease Control and Prevention Agency	Women’s health research for prevention and management of non-communicable diseases	Hyun-young Park	Chungcheongbuk-do	278
	Industry	Wisenut	Development of an interactive medical history taking software based on lifelog data for chronic disease patients	Wooyoung Kwon	Gyeonggi	833
	Industry	Medical Excellence	System advancement and development for chronic disease monitoring and education in primary clinics	Yoon-hee Choi	Seoul	292
	Hospital	Samsung Medical Center	Advancement and demonstration of a primary care-based chronic disease monitoring service model	Jaeheon Kang	Seoul	208
	University	Catholic University of Korea	Development of advanced system linkage service model for the optimal patient care of chronic diseases in primary clinics	Gun-ho Yoon	Seoul	125
	University	University of Washington	Central hub for kidney precision medicine	Jonathan Himmelfarb	Seattle, WA, USA	4286
	University	Academisch Ziekenhuis Groningen	Personalised medicine in diabetic chronic disease management	Hiddo J. L. Heerspink	Groningen, Netherlands	3794
	University	University College London	MICA: Medical Bioinformatics: Data-driven discovery for personalised medicine	Peter Coveney	London, UK	11,685
	University	The University of Tokyo	Development of a diagnostic algorithm through gene panel testing and genetic risk score analysis to facilitate precision medicine for diabetes	Hosoe Jun	Tokyo, Japan	35

## Data Availability

Not applicable.

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
