# Peer review of "Public R&D Projects-Based Investment and Collaboration Framework for an Overarching South Korean National Strategy of Personalized Medicine"

_ijerph, 2022, doi:10.3390/ijerph19031291_

Round 1

Reviewer 1 Report

A good attempt. Please see the attachment

Author Response

Point-by-point responses to reviewers’ comments (detail the revisions)

Thank you for the reviews of our manuscript IJERPH-1536392 entitled "Public R&D Projects-based Investment and Collaboration Framework for an Overarching South Korean National Strategy of Personalized Medicine".

We are grateful for the opportunity to submit a revised version of our manuscript and sincerely thank the reviewers for their constructive criticism and helpful comments that we have used to improve the manuscript.

Below we have answered all the comments made by the reviewers.

Reviewer #1 and Author's Response to the Review Report (Reviewer 1):

Comments and Suggestions for Authors:

The authors investigated the “Public R&D projects based Investment and collaboration framework for the overarching Korean national strategy of personalized medicine.” In view of the importance of health to national development in contemporary times, vis-à-vis the perceived lopsidedness in healthcare provision globally, especially in the developing and less developed countries. The research problem is very significant and welcome.

The authors did a good job and I must commend them for their efforts. However, there are some observations that should be of concern to the authors in order to enhance the quality of the paper and make it suitable for publication.

Point 1: Problem definition: the problem definition is fair.

Point 2: Conclusions: the conclusions seem to tie back to the findings and the contributions of the study are evident.

Response 1 & Response 2:

We sincerely appreciate the reviewer’s comments for her/his expert review and the positive assessment of our manuscripts. The authors are very honored by the opportunity to be published in this journal.

Point 3: Empirical Review: The paper is deficient of the required sound empirical review required of a scholarly publication of this magnitude. This tends to make the research gaps unclear.

Point 4: Theoretical Framework: the authors did not underpinned the study with any theory.

Response 3 & Response 4:

The reviewer raises an important point. We agreed with the reviewer’s comment and thank for this recommendation. According to the reviewer's comment, it was described in detail in "1.1.3. Theories and Empirical Review" of the “Introduction” section through an empirical literature review (please see below) [Page 2, line 63-78; Page 4-5, line 131-210].

(revised manuscript)

1. Introduction

……Thus, recent studies have highlighted governments’ endeavors in achieving the goals of PM through effective collaborations with other stakeholders in the academic, public, private, and health sectors at the international and inter-regional levels [5,17–21]. However, most studies have focused on the normative argument that governments should play a core role in encouraging new and more flexible forms of collaboration networks at the local, national, and international levels [13,22]. Prior research has not contributed to mitigating inherent bias of scientists and experts who participated in the public funding decision making process to set directions of a variety of programs because useful information was not provided. Thus, it is necessary for scientists and experts to provide salient, credible, and legitimate information stemming from a systematic framework for investment and collaborative research strategies of PM, thereby avoiding the potentially biasing influence in decision making [23–26]. To complement the lack of such framework studies to implement strategies, we adopted our framework [27,28], which could present the information about the current status of and future directions for the overarching South Korean national research and development (R&D) strategy of PM.

1.1.3. Theories and Empirical Review

Research-funding organizations often must make difficult decisions about resource allocation [35,36]. Theories regarding decision making generally fall into an information-processing approach that traces back to Herbert Simon and the notion of bounded rationality [35–37]. Essentially, information processing in relation to human decision-making, containing knowledge acquisition and communication among decision-makers, is formed ("bounded") by the limits of individual attention and heuristics [38]. Consequently, considerable skepticism concerning the reliability and legitimacy of peer review [35] and committee and group meetings [40] has risen among certain parts of the scientific world, because scientists and experts who participated in the decision-making have inherent cognitive bias stemming various sources including the specific case (e.g., data, reference materials, and contextual information), environment, culture and experience (e.g., organizational factor, education and training), and human nature (e.g., personal factors, human brain) [26].

As a general principle to mitigate bias, it is necessary to focus solely on the relevant information [23–26, 41]. Furthermore, the information being processed requires three attributes [23]—salience, credibility, and legitimacy—and governments should establish institutional roles for procedures that can provide and exchange timely, coherent, and trustworthy information and for coordination among stakeholders [42].

Recently new techniques such as AI or machine learning, which improve accuracy, consistency, and fairness, have been adopted to mitigate human biases and decrease the inevitable variability in decision-making [25]. Simultaneously, information-shared platforms, such as that provided by the World Health Organization (platform.who.int/data/), have also been introduced to encourage more discussion and interaction with other stakeholders.

For example, the OECD has provided statistics on R&D expenditure of its members for national and international policymakers, and the statistics enabled them to measure who conducts and who funds R&D and where it takes place, the level and purpose of such activities, and interactions and collaborations between institutions and sectors to evaluate or understand how R&D contributes to economic growth/activities from a broad perspective [41,43].

Some of scholars have used the public (R&D) expenditure in part to considerate direction of innovation policies in decision-making. Comparing Indonesian innovation capacity developed by increasing R&D expenditure with Asian countries (Singapore, Malaysia, Thailand, Vietnam, South Korea, and China), Aminullah [44] indicated that the capacity has continued to decline so that the government R&D budget is needed to reach a certain level of GDP. In particular, the acquisition of high technical capability of Indonesia emerged as an urgent policy issue, and it must to be coordinated with the science and technology policy for economic development by an integrated R&D governance body.

By reviewing the characteristics of mission-oriented programs for innovation policy, Mazzucato [45] suggested that one of the important principals of policymaking is to actively set a direction of change with dynamic debates across different sectors and actors in an economy through ensuring democratic legitimacy. In particular, investigating the trend of public investment by mission-oriented institutes like the National Institutes of Health (NIH) may be the starting point as a deliberate and targeted direction for discussion of public R&D funding.

By comparing the allocation of EU research funds among member states, Begg [46] argued that the directions of the allocated proportions of funding should be revised considering criteria associated with the challenges the EU confronted, and the higher levels of territorial cooperation projects are encouraged to mitigate the adverse effects of internal borders and to provide innovative solutions to common problems such as environmental issues, energy, and healthcare.

Regarding PM, few studies have investigated the status of research investments for policy directions. Nimmesgern et al. [6] addressed challenges for advancing PM in the context of the medical innovation cycle, and the major challenge was identified as the use of omics technologies, demanding a significant continuous investment. Meanwhile, they briefly discussed the breadth of PM projects aided through EU research funding from FP7 (2007–2013) and Horizon 2020 (2014–2020) programs to address a broad range of R&D activities from largescale data collection, omics technologies development, rare disease research, and new diagnostic testing to piloting PM in healthcare. These prime examples of PM projects allowed experts in the PM community to present the research area to be investigated and initiate projects tackling the diverse challenges requiring investment. Furthermore, it is critical to invest continuously in collaboration research across Europe and the world for successful implementation of PM.

In line with this argument, many studies on PM have commonly emphasized that governments should have an important role in promoting various forms of collaboration for R&D systems and supply chains throughout multiple connecting domains, stakeholders, and supplies at the local, national, and international levels [18,20,47,48] to solve clinical and operational issues including lack of electronic health record tools, absence of clinical decision support tools, and demand for integration of test results [13,49].

On the other hand, Maughan [50] argued that optimism for PM may lead funding allocation to areas of apparent success to the detriment of other important research areas including diseases associated with poverty and health inequalities in rural areas [38,39]. Thus, the criteria for research funding should follow clinical and population priorities rather than persisting in the apparent promise of the science to further improvement in health and healthcare.

However, these studies on PM have focused on the normative argument without analyzing the trend and/or status of research funding and presenting the evidence-based collaboration approaches. Thus, it is necessary to provide useful R&D funding-related information that may be utilized in decision-making for the direction of PM-related research.

Point 5: Grammar: the authors need to do a thorough editing of the work to get rid of the numerous grammatical errors.

Response 5:

As the reviewer’s comment, the authors desperately realized that the English grammar and expressiveness of our manuscript were quite insufficient, and we sincerely apologize for this. We spent as much time as possible to request professional translation services by native English-speaking experts. We expect that extensive English correction/proofreading throughout the manuscript will improve readability and sufficiently improve the quality of the manuscript.

Point 6: Methods: The authors should elucidate the methodology to give a clearer view to readers and thus make replication of the study easier.

Response 6:

As the reviewer’s comment, we detailed our methods to give readers a clearer view and to make it easier to replicate research. We followed the reviewer’s recommendations and have described the “Method” sections in three separate sections – “2.1. Data collection and preprocessing”, “2.2. Co-occurrence matrix”, “2.3. Clustering and network visualization” – of the revised manuscript according to the reviewer’s recommendation (please see below) [Page 6-10, line 237-321].

(revised manuscript)

2.1. Data Collection and Preprocessing

In this study, we collected PM-related R&D projects using the National Institute of Science and Technology, which holds information on all national R&D projects in South Korea. The title and abstract of national R&D project data were translated into English. Then, with experts from universities, research institutes, industries, and hospitals, the following keywords and their variants were used to determine search strategies and final data: “precision medicine,” “precise medicine,” “personalized medicine,” “personalised medicine,” “personalized therapy,” “personalised therapy,” “individualized medicine,” “individualized therapy,” “individualised medicine,” “individualised therapy,” “tailored medicine,” “tailored therapy,” “customized medicine,” “customized therapy,” “customised medicine,” “customised therapy,” “precision health,” “targeted treat,” “targeted therapy,” “preventive medicine,” “predictive medicine,” “cohort precision medicine,” and “omics technologies.” The dataset is described in Table 1. The data on 8,478 nationally funded PM-related R&D projects from 112 R&D programs between 2015 and 2020 were collected, and the experts examined whether they were associated with PM, bringing the data sample to 5,727 projects. After removing projects with missing investment information, the final dataset contained 5,647 projects with a value of USD 1,408.5 million (Tables 2 and 3). To identify the characteristics of these nationally funded R&D projects, we used the All Science Journal Classification (ASJC) model developed through machine learning employing author keywords from approximately 10,000 recent articles in each ASJC field (e.g., 1311: Genetics, 1312: Molecular Biology) in the Scopus database as the feature and the 344 field names of the ASJC codes used to classify journals in Scopus as the labels [27, 28]. Three ASJC codes (labels) were attached to each public R&D project, and the probability of relevance of each assigned ASJC code was indicated based on information (feature) from the title and abstract of the R&D project. In addition, a 10% threshold probability was applied to improve the correlation between the assigned ASJC codes and the nationally funded R&D projects. Figure 2 shows a conceptual diagram of this process.

2.2. Co-occurrence Matrix

Based on earlier studies [27,51,52], a co-occurrence technique was employed to determine PM-related research fields. The number of simultaneous appearances of ASJC codes in a project group indicated the relevance of that project. Specifically, the number of times element i (from the first list) and element j (from the second list) emerged together in the text was provided by the co-occurrence matrix: i,j = ASJC codes. The more ASJC codes appear in a group, the higher the relevance of the projects that have these ASJC codes.

2.3. Clustering and Network Visualization

The network among projects was built based on the number of appearances of ASJC codes in the projects. All nodes in the network were produced by the titles of the research fields present in the ASJC codes, and the font size implied the frequency of the co-occurrence of each ASJC code compared to other codes. Visualizing the network structure enables researchers to understand the relationship between ASJC codes. This study used the modularity-based clustering method from VOSviewer (Leiden University, Leiden, The Netherlands, Version 1.16.15) for its consistency and accuracy of results [53]. The mapping and clustering were calculated by minimizing Equation (1), which describes the clustering algorithm and is explained in a previous study [52,54].

           (1)

(2)

where n is the number of nodes in the network, m is the number of links in the network, cij is the number of links between nodes i and j, and ci is the number of nodes i.

With respect to , …, ,  is the distance between nodes i and j. For the mapping,  was calculated using the following formula:

              (3)

(4)

where  is a vector denoting the location of node i in a p-dimensional map. For the clustering,  was calculated using the following formula:

                             (5)

(6)

where  = integer denotes the cluster to which node i belongs, γ = resolution parameter.

The numbers of the clustering were decided by the resolution parameter (γ > 0); the higher the value of the parameter, the larger the number of clusters created. The number of clusters ranged from 1 (γ = 0.1) to 8 (γ = 0.9). Considering the number and combination of items (i.e., ASJC codes) in individual clusters and the value chain of PM, eight clusters were finally determined.

2.4. Defining the PM-Related Research Fields

The PM-associated research fields were defined throughout the experts’ discussion process, which considered both the information about title and content of the projects and the distribution of ASJC codes comprising each cluster. This process enabled experts to reach the conclusion using a variety of information, thereby improving the legitimacy of the process and the perception of it as unbiased and fair [23,24]. To identify potential international collaboration research organizations in some research areas, experts searched in some targeted funding databases (i.e., the Community Research and Development Information Service of the European Commission, RePORT of the NIH in the US, the UK Research and Innovation of UK, and the Database of Grants-in-Aid for Scientific Research of Japan). The entire process is detailed in Figure 3.

Figure 3. Process of data collection and analysis of nationally-funded R&D projects related to PM.

Point 7: Limitation: The first limitation ought not to constrain the authors because it is surmountable. Except there is a peculiar institutional deficiency, the authors should be able to access the expenditures of the 18 Local governments if they are determined.

Response 7:

We thank for this recommendation. According to the reviewer’s comment,  we have described the limitation section by adding related content and implications in 5.1. Limitations and further research of the “Conclusions” section of the manuscript (please see below) [Page 22-23, line 618-632].

(revised manuscript)

5.1. Limitations and Further Research

Despite these contributions, our study also posed the same limitations that presented challenging questions for future research [27]. Only the data of public R&D projects from the central government were utilized; due to the absence of a database for the R&D expenditures of the 17 local governments, it is currently impossible to integrate the local government-funded project dataset. Thus, it will be necessary to use the proposed framework to accurately understand the status and trends of PM-related technologies when the local government-funded project dataset was developed and assessed by the public. Despite these restrictions, if a consensus on data openness is formed in 18 local governments in the future and a data-sharing plan is prepared, an analysis of detailed R&D perspectives between the central and local governments based on integrated data can be expected. Another limitation was the lack of funding data from other countries, such as the US, EU, and Japan, which could be employed to conduct a comparative analysis for an international research collaboration network among the technology segments, working toward a successful implementation of PM.

Reviewer 2 Report

The article offers a detailed description of the Korea National Strategy of Personalized Medicine by analyzing the distribution, over topics and location, of 5.727 projects financed between 2015-2020. 

The study is a great source of information but the writing has to be greatly improved.

Just to make some examples:

line 139 "they commonly argued", we don't know to who "they" refers to and the verb tense in wrong, either "argue" or "have argued"

line 163 and line 174: are we worried about health equalities or inequalities? 

line 177: "have been perceived" by whom?

line 180: how do we challenge a challenge? and which one has to be challenged?

These are the examples from a single page and many more can be found in the paper. Extensive proofreading is needed.

Regarding the content of the paper: Although the introduction of the paper sets Korea in the international framework regarding national strategies for personalized medicine an international perspective is lacking in the final discussion. It would be interesting to know how the clusters from Korea compare  with those from other nations, of at least with the objectives identified by the national strategies described in the introduction. 

Author Response

Point-by-point responses to reviewers’ comments (detail the revisions)

Thank you for the reviews of our manuscript IJERPH-1536392 entitled "Public R&D Projects-based Investment and Collaboration Framework for an Overarching South Korean National Strategy of Personalized Medicine".

We are grateful for the opportunity to submit a revised version of our manuscript and sincerely thank the reviewers for their constructive criticism and helpful comments that we have used to improve the manuscript.

Below we have answered all the comments made by the reviewers.

Reviewer #2 and Author's Response to the Review Report (Reviewer 2):

Comments and Suggestions for Authors

The article offers a detailed description of the Korea National Strategy of Personalized Medicine by analyzing the distribution, over topics and location, of 5.727 projects financed between 2015-2020.

Point 1: The study is a great source of information but the writing has to be greatly improved. Just to make some examples. These are the examples from a single page and many more can be found in the paper. Extensive proofreading is needed.

Response 1:

We appreciate the reviewer for her/his sincere criticism and advice. As the reviewer’s comment, the authors desperately realized that the English grammar and expressiveness of our manuscript were quite insufficient, and we sincerely apologize for this. We spent as much time as possible to request professional translation services by native English-speaking experts. We expect that extensive English correction/proofreading throughout the manuscript will improve readability and sufficiently improve the quality of the manuscript.

Point 2: Regarding the content of the paper: Although the introduction of the paper sets Korea in the international framework regarding national strategies for personalized medicine an international perspective is lacking in the final discussion. It would be interesting to know how the clusters from Korea compare with those from other nations, of at least with the objectives identified by the national strategies described in the introduction.

Response 2:

We sincerely appreciate for the reviewer for suggesting this. This comments significantly improved the quality and completeness of our manuscript. In response to reviewer’s comments, we highlight the previously insufficient international perspective and its implications, as well as the original contribution and development of this study in the “Materials and Methods”, “Results”, "Discussion" and "Conclusions" sections (please see below).

(Before)

2. Materials and Methods

2.3. Defining the PM-Related Research Fields

The PM-associated research fields were defined while considered both the title and content of the projects and the portion of ASJC codes. Experts in each research field group have reached to the agreement during evidence-based discussions. The specific entire process is shown in Figure 3.

Figure 3. Process of data collection and analysis of public R&D global projects related to PM.

3. Results

3.2.5. Potential National Collaborative Partners in R&D related to Three Targeted Dis-eases

….Here we provided three examples of targeted diseases such as cancer, brain dis-ease, and chronic disease for PM inter-regional and/or international collaboration. Table 7 shows information on the field of innovation subjects and organizations, the amount of government investment, and the project manager of representative R&D projects for each of the three target diseases.

Table 7. Representative collaborative research organizations from university, research institutes, hospital, and industry in cancer, brain disease, and chronic disease of PM.

Target Disease

Type of Organization

Organization

R&D Title

Project Manager

Region

Funding

(USD

Thousand)

Cancer

Institute

National Cancer Center

Prognostic impact of CT-determined sarcopenia and sarcopenic obesity in older patients with non-small cell lung cancer undergoing chemotherapy

Yoon-jung Jang

Gyeonggi-do

596

University

Yonsei University

Development of an app-based self-management program “HARU” for cancer patients and testing its effectiveness

Kyungmi Jung

Seoul

11

University

Seoul National University

Evaluation of risk for oral diseases in cancer patients in Korea and the National Health Insurance coverage extension

Seo-kyung Han

Seoul

75

University

Yonsei University

Development of prospective cohort and evidence-based management program for colorectal cancer survivors

Seon-ha Ji

Seoul

55

Brain disease

Hospital

Samsung Medical Center

Protocol development and validation of personalized CNS-PNS hybrid rehabilitation therapy for restoration of gait-related neural network in stroke Patients

Yeon-hee Kim

Seoul

155

Hospital

Seoul National University Hospital

Modeling of prognosis prediction for stroke using big data

Byung-Woo Yoon

Seoul

108

Institute

Korea Institute of Science and Technology

Development of customized rehabilitation technology for stroke patients in neural plasticity evaluation and enhancement

In-chan Yoon

Seoul

1,083

University

Sungkyunkwan University

Effect of digital treatment system on upper limb functional recovery and brain plasticity in stroke patients

Wonhyuk Jang

Gyeonggi-do

83

University

Pusan National University

Effect of digital treatment system on upper limb functional recovery and brain plasticity in stroke patients

Yong-il Shin

Busan

83

University

Gachon University

Development of biomarker monitoring system for verification of Korean medicine treatment towards stroke

Young-jun Kim

Gyeonggi

183

Chronic disease

Industry

M2IT

Intelligent diagnosis prescription inquiry service using CDM-based chronic disease data

Wooseop Shin

Seoul

417

Agency

Korea Disease Control and Prevention Agency

Women’s health research for prevention and management of non-communicable diseases

Hyun-young Park

Chungcheongbuk-do

278

Industry

Wisenut

Development of an interactive medical history taking software based on lifelog data for chronic disease patients

Wooyoung Kwon

Gyeonggi

833

Industry

Medical Excellence

System advancement and development for chronic disease monitoring and education in primary clinics

Yoon-hee Choi

Seoul

292

Hospital

Samsung Medical Center

Advancement and demonstration of a primary care-based chronic disease monitoring service model

Jaeheon Kang

Seoul

208

University

Catholic University of Korea

Development of advanced system linkage service model for the optimal patient care of chronic diseases in primary clinics

Gun-ho Yoon

Seoul

125

(After) [Page 9-10, line 308-321; Page 18-20, line 512-519].

2. Materials and Methods

2.4. Defining the PM-Related Research Fields

The PM-associated research fields were defined throughout the experts’ discussion process, which considered both the information about title and content of the projects and the distribution of ASJC codes comprising each cluster. This process enabled experts to reach the conclusion using a variety of information, thereby improving the legitimacy of the process and the perception of it as unbiased and fair [23,24]. To identify potential international collaboration research organizations in some research areas, experts searched in some targeted funding databases (i.e., the Community Research and Development Information Service of the European Commission, RePORT of the NIH in the US, the UK Research and Innovation of UK, and the Database of Grants-in-Aid for Scientific Research of Japan).

Figure 3. Process of data collection and analysis of nationally-funded R&D projects related to PM.

3. Results

3.2.5. Potential National Collaborative Partners in R&D related to Three Targeted Dis-eases

….Here we provided three examples of targeted diseases such as cancer, brain dis-ease, and chronic disease for PM inter-regional and/or international collaboration. Table 7 shows information on the field of innovation subjects and organizations, the amount of government investment, and the project manager of representative R&D projects for each of the three target diseases.

Table 7. Representative collaborative research organizations from university, research institutes, hospital, and industry in cancer, brain disease, and chronic disease of PM.

Target Disease

Type of Organization

Organization

R&D Title

Project Manager

Region

Funding

(USD

Thousand)

Cancer

Institute

National Cancer Center

Prognostic impact of CT-determined sarcopenia and sarcopenic obesity in older patients with non-small cell lung cancer undergoing chemotherapy

Yoon-jung Jang

Gyeonggi-do

596

University

Yonsei University

Development of an app-based self-management program “HARU” for cancer patients and testing its effectiveness

Kyungmi Jung

Seoul

11

University

Seoul National University

Evaluation of risk for oral diseases in cancer patients in Korea and the National Health Insurance coverage extension

Seo-kyung Han

Seoul

75

University

Yonsei University

Development of prospective cohort and evidence-based management program for colorectal cancer survivors

Seon-ha Ji

Seoul

55

Institute

Broad Institute Inc.

Making cancer precision medicine real bottlenecks and opportunities

Todd R. Golub

Cambridge, MA, USA

1024

University

Royal College of Surgeons in Ireland

Advancing a precision medicine paradigm in metastatic colorectal cancer systems-based patient stratification solutions

Annette Byrne Phd

Dublin, Ireland

6794

University

Queen Mary University of London

Optimal screening and surveillance regimes for early diagnosis of cancer and precision medicine using mathematical modelling

Kit Curtius

London, UK

370

University

Keio University

Establishment of small cell lung cancer organoids for development of precision medicine

Mitsuishi Akifumi

Tokyo, Japan

37

Brain disease

Hospital

Samsung Medical Center

Protocol development and validation of personalized CNS-PNS hybrid rehabilitation therapy for restoration of gait-related neural network in stroke Patients

Yeon-hee Kim

Seoul

155

Hospital

Seoul National University Hospital

Modeling of prognosis prediction for stroke using big data

Byung-Woo Yoon

Seoul

108

Institute

Korea Institute of Science and Technology

Development of customized rehabilitation technology for stroke patients in neural plasticity evaluation and enhancement

In-chan Yoon

Seoul

1,083

University

Sungkyunkwan University

Effect of digital treatment system on upper limb functional recovery and brain plasticity in stroke patients

Wonhyuk Jang

Gyeonggi-do

83

University

Pusan National University

Effect of digital treatment system on upper limb functional recovery and brain plasticity in stroke patients

Yong-il Shin

Busan

83

University

Gachon University

Development of biomarker monitoring system for verification of Korean medicine treatment towards stroke

Young-jun Kim

Gyeonggi

183

University

Ohio State University

Laying the groundwork for personalized medicine in aphasia therapy genetic and cognitive predictors of restorative treatment response

Stacy M. Harnish

Columbus, Ohio, USA

487

University

Charité-Universitätsmedizin Berlin

Personalised medicine by predictive modeling in stroke for better quality of life

Dietmar Frey

Berlin, Germany

6773

University

King's College London

Towards personalised medicine in psychiatric genetics the role of cardiometabolic traits in severe mental illness

Saskia Hagenaars

London, UK

409

University

Hamamatsu University School of Medicine

Precision medicine in developmental psychiatry

Kenji J. Tsuchiya

Shizuoka, Japan

159

Chronic disease

Industry

M2IT

Intelligent diagnosis prescription inquiry service using CDM-based chronic disease data

Wooseop Shin

Seoul

417

Agency

Korea Disease Control and Prevention Agency

Women’s health research for prevention and management of non-communicable diseases

Hyun-young Park

Chungcheongbuk-do

278

Industry

Wisenut

Development of an interactive medical history taking software based on lifelog data for chronic disease patients

Wooyoung Kwon

Gyeonggi

833

Industry

Medical Excellence

System advancement and development for chronic disease monitoring and education in primary clinics

Yoon-hee Choi

Seoul

292

Hospital

Samsung Medical Center

Advancement and demonstration of a primary care-based chronic disease monitoring service model

Jaeheon Kang

Seoul

208

University

Catholic University of Korea

Development of advanced system linkage service model for the optimal patient care of chronic diseases in primary clinics

Gun-ho Yoon

Seoul

125

University

University of Washington

Central hub for kidney precision medicine

Jonathan Himmelfarb

Seattle, WA, USA

4286

University

Academisch Ziekenhuis Groningen

Personalised medicine in diabetic chronic disease management

Hiddo J. L. Heerspink

Groningen, Netherlands

3794

University

University College London

MICA: Medical Bioinformatics: Data-driven discovery for personalised medicine

Peter Coveney

London, UK

11685

University

The University of Tokyo

Development of a diagnostic algorithm through gene panel testing and genetic risk score analysis to facilitate precision medicine for diabetes

Hosoe Jun

Tokyo, Japan

35

(Before)

4. Discussion

4.1. Discussion for Collaborative overarching R&D Strategy on PM.

The framework proposed in this study is intended to be used as a tool to implement the goal of implementing a successful PM and the goal of improving regional equality in public health and health care in three aspects: region, technology and organization. The following three important research results were obtained through six research questions to prove the usefulness of the framework. First, the government's PM investment status and R&D trends conducted over the past five years were examined, focusing on RQ1-1 and RQ1-2. This information is provided to policy makers to present the direction of public R&D and contributes to rationalizing decision-making on the selection and concentration of each R&D area. Second, in order to grasp the technological competitiveness of the PM field from a regional perspective through RQ2-1 and RQ2-2, trends according to the regional distribution of the PM-related technology field were examined. Central and regional stakeholders can discuss collaboration to strengthen mutual competitiveness, focusing on regions with high innovation in each research field by using PM's investment information in 8 and 17 regions as a medium. Finally, focusing on RQ3-1 and RQ3-2, major innovative organizations in each region divided into universities, industries, research institutes, and hospitals, and their R&D activities and R&D areas were examined. This information can be used as the basis for mutual cooperation R&D partnerships between regions based on the R&D potential and technological competitiveness of R&D projects.In order for the government to establish policies for PM and present implementation strategies accordingly, it must be preceded by discovering innovative organizations with technology development competitiveness and diagnosing R&D portfolios by region by specific industry. In order to secure technological competitiveness and establish a cooperative strategy with innovative institutions across the country to provide high-quality medical and health care services to the aging population in rural areas. Therefore, this framework can be used as an empirical analysis tool to revitalize collaborative R&D between the central and local governments, and provides necessary information as a starting point for R&D support policy to promote balanced development by fostering specialized industries and strengthen public health. This framework is not only practical for R&D innovation organizations in each region to discover various cooperative partners across the country, but can also contribute to discovering opportunities in new R&D and business areas with them and creating value by expanding their roles and competitiveness. For example, universities and research institutes can discover new research opportunities, expand science and technology infrastructure, present directions for education and training for employment and commercialization, and hospitals can ultimately implement personalized medical care and find innovative and useful business benefits.

(After) [Page 20-21, line 520-563].

4. Discussion

4.1. Discussion for Collaborative Overarching R&D Strategy on PM

The proposed framework for an overarching national collaborative R&D strategy for PM provides a variety of information to improve the directions of experts’ funding decision-making towards realizing the successful implementation of PM and improving regional equality for public medical and healthcare in terms of regional, technological, and organizational dimensions. To demonstrate the utilization of the framework, we established six research questions.

First, the status and trends of government funding in PM-related technologies during 2015–2020 were examined based on RQ1-1 and RQ1-2. This information can allow experts and/or stakeholders to discuss the appropriateness of national R&D investment while diminishing human biases and reducing the inevitable variability in the decision-making process.

Second, the distribution and trends of R&D funding in PM-related technology areas during 2015–2020 were investigated to grasp the technological competitiveness of the PM field from a regional perspective through RQ2-1 and RQ2-2. South Korean central and regional stakeholders can discuss developing collaboration programs to strengthen mutual competitiveness, focusing on regions with high innovation capability in each research field by using PM’s investment information in terms of both eight research fields and 17 regions as a medium.

Finally, focusing on RQ3-1 and RQ3-2, major innovative organizations in each region were divided into universities, industries, research institutes, and hospitals, and their R&D activities and R&D areas were examined. This information can be used as the basis for collaborative R&D partnerships in technology areas at the national and international levels because it contains the objective contents of R&D projects such as research title, project manager, funding scale, and region.

To review the directions for establishing comprehensive national collaboration and implementation policies of PM while mitigating cognitive bias of stakeholders who participated in decision-making process, it should be preceded by creating information about the status of innovative organizations with technology development competitiveness and diagnosis of R&D portfolios of regions in the industry. This may allow the governments to provide better medical and healthcare services to the aging population in rural areas. Therefore, this framework can be used as an empirical analysis tool to revitalize collaborative R&D between central and local governments and provides necessary information as a starting point for R&D support policy to promote balanced development by fostering specialized industries and strengthen public health.

This framework not only is practical for R&D innovation organizations in each region to discover various cooperative partners across the country but also can contribute to discovering opportunities in new R&D and business areas and creating value by expanding their roles and competitiveness. For example, universities and research institutes can discover new research opportunities, expand science and technology infrastructure, and present directions for education and training for employment and commercialization, and hospitals can ultimately implement personalized medical care and find innovative and useful business benefits.

(Before)

5. Conclusions

Since the Korea government designated PM as a national strategic task in 2016, it has continued to make investments to achieve its goals that have recently accelerated due to the COVID-19. Furthermore, long-term efforts among academia, public, private and health sectors are required for successful PM implementation [26,30]. This requires a fine-tuned investment analysis framework that reflects regional variations due to disparities in relevant assets such as human resources, market size, and institutional factors [22]. This study, which began with our previous work, empirically demonstrated that the framework can present accurate cross-regional innovation plans, taking into account regional, technical and organizational dimensions to build horizontal and vertical collaborations between different stakeholders, actors networks and policy actors in different spaces. It has been shown empirically that it can be done. Therefore, this study makes two important contributions. First, we present an investment and collaboration framework that provides information on the status and trends of government R&D investments in the technology sector in the target area. Based on previous studies [22,23], data on national research projects, project period, and funding data were used. Thus, this information enables the PM's R&D managers to form strategic collaborations while taking into account the limited resources, lack of knowledge, and uncertainty about the market in the early PM industry.

The second is that it has proven how to utilize the framework for PM. A comparative analysis of investment levels in government-funded research projects related to PM for each region and other technology clusters in Korea during 2015-2020, and various R&D institutions in each cluster and region are described. By explicitly presenting innovation capabilities in eight distinct technology areas across 17 regions, it not only shows regional differences in individual areas, but also proposes a list of organizations of PM services such as cancer, brain diseases and chronic diseases by region. Seoul has the highest technological prowess compared to other regions, and some regions have relatively superior competencies in specific technological areas, such as Daegu's clinical information, Gangwon-do's service, and Ulsan's cohort. These results show empirical evidence for the differentiation of regional competitiveness and promote the discovery of inter-regional cooperative R&D partners for better rural-urban public health services.

5.1. Limitations and Further Research.

Despite these contributions, our study also posed the same limitations that presented challenging questions for future research in the previous study [22]. Only the data of public R&D projects from the central government were utilized; the R&D expenditures of the 18 local governments are required in the proposed framework to accurately understand the status and trends of PM-related technologies. Another limitation was the lack of funding data from other countries, such as the US, EU, and Japan, which could be employed to conduct a comparative analysis for an international research collaboration network among the technology segments, working toward a successful implementation of PM.

(After) [Page 21-23, line 564-632].

5. Conclusions

Since the South Korean government designated PM as a national strategic task in 2016, it has continued to make investments to achieve its goals, which were recently accelerated due to the COVID-19 pandemic. Furthermore, long-term efforts among academia, public, and private and health sectors are required for successful PM implementation [31]. This requires a fine-tuned investment analysis framework that reflects regional variations due to disparities in relevant assets such as human resources, market size, and institutional factors [27].

This study, which began with our previous work, empirically demonstrated that the framework can present accurate cross-regional innovation plans, considering regional, technical, and organizational dimensions to build horizontal and vertical collaborations between different stakeholders, actor networks, and policy actors in different spaces. Therefore, this study makes four important contributions. First, we present an investment and collaboration framework that provides information on the status and trends of government R&D investments in the technology sector in the target area. Based on previous studies [27,28], data on national research projects, project period, and funding were used. Thus, this information enables the PM’s R&D managers to form strategic collaborations while considering the limited resources, lack of knowledge, and uncertainty about the market in the early PM industry.

The second contribution is that this study shows how to utilize the framework for PM. It provides a comparative analysis of investment levels in government-funded research projects related to PM for each region and other technology clusters in South Korea during 2015–2020 and describes various R&D institutions in each cluster and region. By explicitly presenting innovation capabilities in eight distinct technology areas across 17 regions, it not only shows regional differences in individual areas but also proposes a list of organizations of PM services such as cancer, brain diseases, and chronic diseases by region. Seoul has the highest technological prowess compared to other regions, and some regions have superior competencies in specific technological areas, such as Daegu's clinical information, Gangwon-do's service, and Ulsan's cohort. These results show empirical evidence for the differentiation of regional competitiveness and promote the discovery of inter-regional cooperative R&D partners for better rural–urban public health services.

Thirdly, COVID-19 not only highlighted the importance of the global common agenda in the medical system but also awakened the international community's efforts centered on science and technology cooperation are absolutely essential [14,51]. Based on South Korea’s personalized medical technology capabilities and experiences, the framework presented in this study provides basic information to establish and promote differentiated strategies for technology development cooperation with advanced countries according to similar target diseases, purposes, and functions. In the future, it is expected that South Korea will play a role in practical international cooperation through technical support and cooperation networks with developing countries centered on South Korea’s PM strengths and R&D base organizations.

Finally, many studies [35–37] pointed out that the cognitive bias of experts stemming from the limits of individual attention and heuristics incurred skepticism concerning the reliability and legitimacy of the decision-making, requiring a procedure that can provide useful information for coordinating stakeholders while mitigating the bias and decreasing variability in decision-making. The proposed framework contributed to tackling the gap between theories and practices throughout, providing information that has salience, credibility, and legitimacy [23,24]. Especially, this study can reduce the practical barriers posed by lack of information allowed to debate the direction of public R&D funding in the decision-making process, issued by other scholars [6,45,50]. At the same time, the insufficiency of the normative argument that governments’ core role in forming PM-related collaboration networks at the local, national, and international levels emphasized in previous studies [2–6,10,12,13,19,22] can be improved by this study.

5.1. Limitations and Further Research

Despite these contributions, our study also posed the same limitations that presented challenging questions for future research [27]. Only the data of public R&D projects from the central government were utilized; due to the absence of a database for the R&D expenditures of the 17 local governments, it is currently impossible to integrate the local government-funded project dataset. Thus, it will be necessary to use the proposed framework to accurately understand the status and trends of PM-related technologies when the local government-funded project dataset was developed and assessed by the public. Despite these restrictions, if a consensus on data openness is formed in 18 local governments in the future and a data-sharing plan is prepared, an analysis of detailed R&D perspectives between the central and local governments based on integrated data can be expected. Another limitation was the lack of funding data from other countries, such as the US, EU, and Japan, which could be employed to conduct a comparative analysis for an international research collaboration network among the technology segments, working toward a successful implementation of PM.

Round 2

Reviewer 2 Report

Dear authors,

thank you for adding some informations on the international dimension.

I think that the English still needs some improvements.

Minor point: Table 7 reports twice a project regarding the "Effect of digital treatment system on upper limb functional recovery and brain plasticity in stroke patients". Is this correct?

Author Response

Reviewer #2 and Author's Response to the Review Report (Reviewer 2):

Comments and Suggestions for Authors

Minor point: Table 7 reports twice a project regarding the "Effect of digital treatment system on upper limb functional recovery and brain plasticity in stroke patients". Is this correct?.

Response 1:

We thank for the reviewer’s careful review. Two projects regarding the "Effect of the digital treatment system on the recovery of upper limb function and brain plasticity of stroke patients" are a consortium project consisting of two universities. Therefore, according to the reviewer's comment, the authors decided to present only the host university as a representative project (please see below).

Table 7. Representative collaborative research organizations from university, research institutes, hospital, and industry in cancer, brain disease, and chronic disease of PM.

Target Disease

Type of Organization

Organization

R&D Title

Project Manager

Region

Funding

(USD

Thousand)

Brain disease

Hospital

Samsung Medical Center

Protocol development and validation of personalized CNS-PNS hybrid rehabilitation therapy for restoration of gait-related neural network in stroke Patients

Yeon-hee Kim

Seoul

155

Hospital

Seoul National University Hospital

Modeling of prognosis prediction for stroke using big data

Byung-Woo Yoon

Seoul

108

Institute

Korea Institute of Science and Technology

Development of customized rehabilitation technology for stroke patients in neural plasticity evaluation and enhancement

In-chan Yoon

Seoul

1,083

University

Sungkyunkwan University

Effect of digital treatment system on upper limb functional recovery and brain plasticity in stroke patients

Wonhyuk Jang

Gyeonggi-do

83

University

Pusan National University

Effect of digital treatment system on upper limb functional recovery and brain plasticity in stroke patients

Yong-il Shin

Busan

83

University

Gachon University

Development of biomarker monitoring system for verification of Korean medicine treatment towards stroke

Young-jun Kim

Gyeonggi

183

University

Ohio State University

Laying the groundwork for personalized medicine in aphasia therapy genetic and cognitive predictors of restorative treatment response

Stacy M. Harnish

Columbus, Ohio, USA

487

University

Charité-Universitätsmedizin Berlin

Personalised medicine by predictive modeling in stroke for better quality of life

Dietmar Frey

Berlin, Germany

6773

University

King's College London

Towards personalised medicine in psychiatric genetics the role of cardiometabolic traits in severe mental illness

Saskia Hagenaars

London, UK

409

University

Hamamatsu University School of Medicine

Precision medicine in developmental psychiatry

Kenji J. Tsuchiya

Shizuoka, Japan

159
